# Integration host factor regulates colonization factors in the bee gut symbiont *Frischella perrara*

Konstantin Schmidt[1†‡], Gonçalo Santos-Matos[1†], Stefan Leopold-Messer[2], Yassine El Chazli[1], Olivier Emery[1], Théodora Steiner[1], Joern Piel[2], Philipp Engel[1*]

[1]Department of Fundamental Microbiology, University of Lausanne, Lausanne, Switzerland; [2]Institute of Microbiology, Eidgenössische Technische Hochschule (ETH) Zurich, Zurich, Switzerland

*For correspondence:
philipp.engel@unil.ch

[†]These authors contributed equally to this work

Present address: [‡]Roche Diagnostics International Ltd,, Rotkreuz, Switzerland

**Abstract** Bacteria colonize specific niches in the animal gut. However, the genetic basis of these associations is often unclear. The proteobacterium *Frischella perrara* is a widely distributed gut symbiont of honey bees. It colonizes a specific niche in the hindgut and causes a characteristic melanization response. Genetic determinants required for the establishment of this association, or its relevance for the host, are unknown. Here, we independently isolated three point mutations in genes encoding the DNA-binding protein integration host factor (IHF) in *F. perrara*. These mutants abolished the production of an aryl polyene metabolite causing the yellow colony morphotype of *F. perrara*. Inoculation of microbiota-free bees with one of the mutants drastically decreased gut colonization of *F. perrara*. Using RNAseq, we found that IHF affects the expression of potential colonization factors, including genes for adhesion (type 4 pili), interbacterial competition (type 6 secretion systems), and secondary metabolite production (colibactin and aryl polyene biosynthesis). Gene deletions of these components revealed different colonization defects depending on the presence of other bee gut bacteria. Interestingly, one of the T6SS mutants did not induce the scab phenotype anymore despite colonizing at high levels, suggesting an unexpected role in bacteria-host interaction. IHF is conserved across many bacteria and may also regulate host colonization in other animal symbionts.

## Editor's evaluation

This fundamental work substantially advances our understanding of the genetic basis of how a very prevalent bee symbiont, Frischella perrara, colonizes the gut of these insects, by identifying novel players in this process and raising new questions related to their mode of action. The authors characterized spontaneous mutants in an important regulator, and showed that this regulator controls the expression of several genes required for gut colonization, they constructed deletion mutants on these genes and characterized these mutants both in vitro and in colonization assays in the presence and absence of other gut symbionts, and provide insights on the mode of action of the novel players identified during host colonization. The combination of approaches used is exceptional and established new standards in the field of host-microbe interactions aiming to understand the molecular players involved in the colonization of gut symbionts.

## Introduction

The digestive tract of many animals is colonized by specialized gut symbionts that occupy distinct physical niches and utilize diverse nutrients. Owing to the availability of genetic tools, gnotobiotic

animal models, and multi-omics approaches, we can now study the genetic features that allow gut symbionts to colonize various animal hosts, including mammals, fishes, and insects (*Wu et al., 2015*; *Lee et al., 2013*; *Townsend et al., 2019*; *Townsend et al., 2020*; *David et al., 2014*; *Nakajima et al., 2018*; *Powell et al., 2016a*; *Ohbayashi et al., 2015*; *Salem et al., 2017*; *Kikuchi et al., 2020*).

The Western honey bee, *Apis mellifera,* is a particularly interesting model to characterize colonization factors of bacterial gut symbionts due to its agricultural importance and the tractability of its gut microbiota. Honey bees harbor relatively simple yet highly specialized gut microbiota composed of 8–10 bacterial genera (*Kwong and Moran, 2016*; *Bonilla-Rosso and Engel, 2018*). The wide distribution of these communities across social bees suggests long evolutionary associations with the host (*Kwong et al., 2017*). Moreover, different members of the bee microbiota colonize distinct physical niches along the gut. Lactobacilli and Bifidobacteria predominate in the posterior hindgut (rectum), while Gammaproteobacteria (*Frischella perrara* and *Gilliamella* species) and a Betaproteobacterium (*Snodgrassella alvi*) preferentially colonize the anterior hindgut (ileum and adjacent pylorus, i.e., the transition zone between midgut and ileum) (*Kwong and Moran, 2016*; *Engel et al., 2015a*; *Martinson et al., 2011*; *Powell et al., 2014*). The partitioning of these bacteria into distinct gut compartments suggests the existence of specific bacterial and/or host mechanisms that facilitate colonization.

Most bacteria of the bee gut microbiota can be cultured and experiments with gnotobiotic bees have been established (*Emery et al., 2017*; *Raymann et al., 2018*; *Zheng et al., 2017*; *Brochet et al., 2021*; *Kešnerová et al., 2017*). Moreover, genomic analyses have provided important insights about the functional potential of bee gut symbionts and their adaptation to the gut environment (*Brochet et al., 2021*; *Ellegaard et al., 2019*; *Ellegaard and Engel, 2019*; *Engel et al., 2012*; *Kwong et al., 2014*; *Ludvigsen et al., 2018*; *Steele et al., 2017*; *Zheng et al., 2019*). Yet, little is known about which genes are directly involved in establishing colonization in the bee gut and how these genes are regulated. The only symbiont that has been extensively studied in this respect is *S. alvi*. Using transposon sequencing and transcriptome analysis, Powell et al. determined genome-wide host colonization factors in *S. alvi* (*Powell et al., 2016a*). Most genes with strong fitness effects were found to belong to three major categories: extracellular interactions, metabolism, and stress response. In particular, genes for attachment and biofilm formation were highly beneficial for colonization, which is in agreement with the observation that this gut symbiont adheres to the host epithelium of the ileum and forms a multispecies biofilm with *Gilliamella*.

The honey bee gut symbiont *F. perrara* belongs to the recently described family Orbaceae within the Gammaproteobacteria (*Volkmann et al., 2010*). It is taxonomically close to the bee gut symbiont *Gilliamella* and has a similar metabolism (*Engel et al., 2013*). However, compared to the other members of the bee gut microbiota, *F. perrara* shows a rather distinctive colonization phenotype. While *S. alvi* and *Gilliamella* have been reported to colonize the host epithelium of the entire ileum (*Martinson et al., 2012*; *Leonard et al., 2018*), *F. perrara* preferentially colonizes the transition zone between the midgut and the ileum, that is, the pylorus. Moreover, colonization with *F. perrara* leads to the appearance of a brown to black material on the luminal side of the epithelial surface between the cuticle layer of the host tissue and the adherent *F. perrara* cells (*Engel et al., 2015a*). This so-called scab phenotype forms after 5–7 d post-colonization and has so far not been reported to be triggered by any other gut symbiont than *F. perrara*. Transcriptome analysis of the host showed that *F. perrara* elicits a specific immune response that includes the upregulation of the host melanization pathway likely responsible for the formation of the scab phenotype (*Emery et al., 2017*). *F. perrara* is highly prevalent across worker bees and colonies of *A. mellifera* (*Engel et al., 2015a*; *Kešnerová et al., 2020*), and related bacteria have also been found in *Apis cerana* (*Wolter et al., 2019*). Moreover, between 25 and 80% of all worker bees of a colony harbor a visible scab phenotype in the pylorus region of the gut, which has been shown to strongly correlate with a high abundance of *F. perrara* (*Engel et al., 2015a*). However, the impact of these phenotype on the host has remained elusive.

Genome sequencing of the type strain of *F. perrara* and comparison with other genomes of the Orbaceae family revealed the presence of several genomic islands that may be involved in the specific interaction of *F. perrara* with the host (*Engel et al., 2015b*). These include a biosynthetic gene cluster for the production of the genotoxic metabolite colibactin (Clb), two distinct type VI secretion systems (T6SSs) and associated effector proteins, type I secretion systems, and fimbrial low-molecular-weight protein (Flp) pili genes. However, currently no genetic tools are available for *F. perrara* precluding

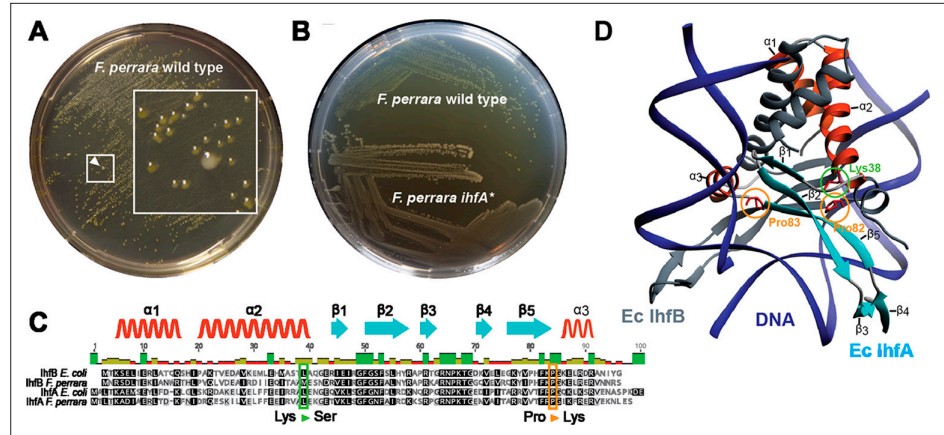

**Figure 1.** Isolation of a spontaneous *ihfA* mutant of *F. perrara* displaying an altered colony morphotype. (**A**) Colonies of *F. perrara* PEB0191 (wt) after 48 hr of growth on modified tryptone yeast glucose (mTYG) agar. Arrowhead points at a larger white colony in between many yellow colonies. The area in the white square is magnified. (**B**) Colony morphology of *F. perrara* wt and the isolated white *ihfA* mutant after growth on mTYG for 48 hr. (**C**) Protein sequence comparison of IhfA and IhfB of *F. perrara* wt and *E. coli* wt. The outlined positions refer to the residues mutated in the three spontaneous *ihfA* mutants: (i) lysine (Lys) to serine (Ser) at position 38 of *F. perrara* IhfA, (ii) proline (Pro) to lysine (Lys) at position 83 of *F. perrara* IhfA, and (iii) proline (Pro) to lysine (Lys) at position 82 of *F. perrara* IhfB. Note that the numbers given on top of the alignment refer to alignment positions and not to positions in the individual sequences. Secondary structures are depicted above as ribbons (α-helix) and arrows (β-sheet) and are numbered according to their appearance in the protein and the structure shown in (**D**). (**D**) Three-dimensional structure of *E. coli* IhfA/B heterocomplex with DNA (source protein databank NDB: PDT040). DNA is depicted in blue and IhfB in dark gray. IhfA is colored according to secondary structure: α-helix orange, β-sheet light blue, and the rest in light gray. α-helices and β-sheets are numbered. The mutated Pro83 and Lys38 residues of *F. perrara* IhfA and the Pro82 residue of IhfB are marked with an orange and green circle, respectively.

The online version of this article includes the following source data and figure supplement(s) for figure 1:

**Figure supplement 1.** Colony morphology of different *F. perrara* strains on modified tryptone yeast glucose (mTYG) agar plates.

**Figure supplement 2.** In vitro characterization of *F. perrara* ihfA*.

**Figure supplement 2—source data 1.** Numeric data underlying the results shown in *Figure 1—figure supplement 2*.

studies about the role of these genetic factors in gut colonization or the induction of the scab phenotype.

Here, we report the isolation of a spontaneous mutant of *F. perrara* that possesses a strong colonization defect in vivo. Resequencing of the mutant revealed a single nonsynonymous point mutation in the gene encoding the alpha subunit of the DNA-binding protein integration host factor (IHF). Using a combination of gnotobiotic bee experiments, transcriptomics, and metabolite analyses, we characterized the genes regulated by IHF. We then established a gene-deletion strategy for *F. perrara*, which allowed us to knock out some of the IHF-regulated genes and show that they impact gut colonization and scab development to different extent in the presence and absence of a complex community.

## Results
### Isolation of spontaneous IHF mutants affecting growth and colony morphology of *F. perrara*

Culturing *F. perrara* type strain PEB0191 (*Engel et al., 2013*) on modified tryptone yeast glucose (mTYG) agar resulted in the formation of yellow colonies. However, we occasionally observed the appearance of larger white colonies among the yellow ones (*Figure 1A*). Restreaking white colonies on fresh mTYG agar usually resulted in yellow colonies again. However, three white colonies that we identified in independent experiments did not change their appearance anymore, suggesting that we had isolated stable 'white' variants of *F. perrara* PEB0191 (*Figure 1B*, *Figure 1—figure supplement 1*).

Genome sequencing of the white variants revealed the presence of three different non-synonymous point mutations in the genes encoding the IHF. IHF is a widely distributed DNA-binding protein consisting of the IhfA/B heterocomplex (*Friedman, 1988*; *Freundlich et al., 1992*; *Goosen and van de Putte, 1995*). Strikingly, two point mutations were identical to each other, but occurred in the different subunits of IHF (*ihfA* and *ihfB*), resulting in a proline to lysine change at amino acid positions 82 and 83, respectively (*Figure 1C*). The third point mutation resulted in a lysine to serine change at position 38 of IhfA (*Figure 1C*). Homology modeling showed that these amino acids are located in the region interacting with DNA, suggesting that the three mutations impact the DNA-binding properties of IHF (*Figure 1D*; *Rice et al., 1996*). As two of the isolated mutants occurred when generating gene deletions of *F. perrara*, they harbored additional genetic modifications (see 'Materials and methods'). Therefore, we focused further characterization of IHF on the mutation Pro83Lys (hereafter *ihfA\**) that occurred in the wild type (wt) background of *F. perrara* PEB0191. While the *ihfA\** strain consistently formed larger colonies than the wt strain on mTYG agar (*Figure 1A and B*), there was no significant difference in growth in liquid culture (permutation test, p=0.097; *Figure 1—figure supplement 2A*). However, light microscopy showed that cells of the mutant strain were on average slightly longer than cells of the wt (Kolgomorov–Smirnov test p<0.0001, *Figure 1—figure supplement 2B and C*).

## *F. perrara* produces an aryl polyene secondary metabolite that is responsible for the yellow colony morphotype

*F. perrara* PEB0191 encodes a genomic island that is homologous to aryl polyene (APE) biosynthetic gene clusters present in other Gammaproteobacteria (*Figure 2A*; *Cimermancic et al., 2014*). APEs are polyunsaturated carboxylic acids conferring a yellow pigmentation to bacterial cells (*Goel et al., 2002*; *Poplawsky et al., 2000*). To assess whether *F. perrara* wt, but not the *ihfA\** mutant, produces an APE, we analyzed cell extracts of both strains by liquid chromatography coupled to heated electrospray ionization high-resolution mass spectrometry (HPLC-HES-HRMS). The data revealed a strongly UV-Vis-absorbent ion peak at $m/z$ 323.1647 [M+H]$^+$, which had a suggested molecular formula of $C_{21}H_{23}O_3$ (*Figure 2B and C*). In the *ihfA\** mutant, this ion was only present at trace amounts (*Figure 2C*). To characterize the metabolite in greater detail, a larger pellet of *F. perrara* wt cultures was extracted and purified by several HPLC runs. Mass spectrometry- (MS) and UV-Vis-guided fractionation yielded an enriched extract that was analyzed by nuclear magnetic resonance (NMR) spectroscopy. The characteristic ions detected in MS-MS fragmentation experiments (*Figure 2—figure supplement 1*), the UV-Vis spectrum with an absorption maximum at 415 nm (*Figure 2—figure supplement 2*) in conjunction with NMR data (*Figure 2—figure supplements 3–10*), suggest an aryl polyene structure identical to that reported in *Cimermancic et al., 2014*; *Schöner et al., 2016*; *Figure 2D*. Unfortunately, it was not possible to connect the NMR substructures because the central methines could not be assigned to chemical shifts (*Figure 2D*, *Figure 2—figure supplement 10*). Comparison of the organic extracts of *F. perrara* wt and *Escherichia coli* CFT073 provided further evidence that both produce the same compound (*Figure 2—figure supplement 11*). Combined, these results suggest that the APE pathway is responsible for the yellow color of the wt colonies of *F. perrara* and is suppressed in the *ihfA\** mutant.

## The *ihfA\** mutant of *F. perrara* has a colonization defect and does not cause the scab phenotype

As APEs have been shown to increase protection from oxidative stress and contribute to biofilm formation (*Cimermancic et al., 2014*; *Schöner et al., 2016*), we sought to test whether the *ihfA\** mutation impacts bee gut colonization. We mono-associated microbiota-free bees with either *F. perrara* wt or *ihfA\**. Colonization with the wt strain resulted in a visible scab in 50 and 80% of all bees after 5 and 10 d of colonization, respectively (n = 18 and n = 36 for both treatments for day 5 and day 10, respectively, *Figure 3A and B*). In contrast, none of the bees colonized with the *ihfA\** mutant developed a visible scab phenotype. To determine whether this difference was due to a general colonization defect of *ihfA\**, we quantified the colonization levels of *F. perrara* at day 5 and day 10 post colonization using colony-forming units (CFUs). While there was a trend towards lower colonization levels (fewer CFUs and more bees without detectable colonization) for the *ihfA\** mutant at day 5 post colonization, the difference was not statistically significant (*Figure 3C*, Wilcoxon rank-sum test p-value = 0.076). However, at day 10 post-colonization, bees colonized with the wt strain showed significantly higher

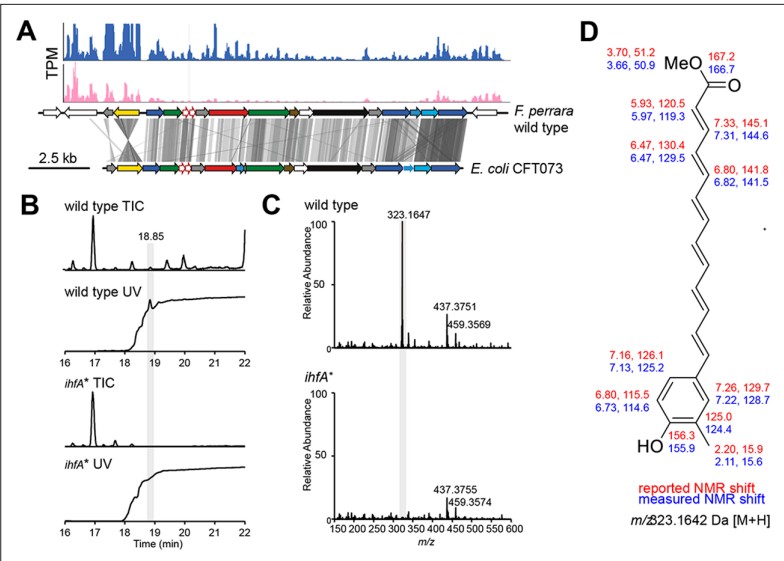

**Figure 2.** Metabolite analysis of *F. perrara* wt and the *ihfA** mutant. (**A**) Comparison of gene synteny and sequence similarity of the genomic islands of *F. perrara* PEB0191 (top) and *E. coli* CFT073 (bottom) encoding the aryl polyene (APE) biosynthesis genes. Gray lines indicate homologous regions based on tblastx analysis. Plots were generated with genoplotR (*Guy et al., 2010*). Transcripts per million (TPMs) are shown on top of the genomic island for one RNAseq replicate of each *F. perrara* wt (blue) and the *ihfA** mutant grown in vitro. Coverage plots were generated with the Integrated Genome Browser v9 (*Freese et al., 2016*). (**B**) Total ion chromatogram (TIC) and UV trace ($\lambda$ = 420 nm) of wt and *ihfA**. A peak highly abundant in the wt was discovered at 18.85 min. Its high UV absorbance at $\lambda$ = 420 nm indicated a conjugated carbon double bond system. (**C**) The normalized mass spectrum at 18.85 min reveals the ion m/z = 323.1647 Da to be approximately 50-fold more abundant in the wt compared to *ihfA**.
(**D**) Enrichment of the ion containing fraction by HPLC followed by nuclear magnetic resonance (NMR) experiments suggest a structure identical to that reported by *Cimermancic et al., 2014*. Reported (red) and observed (blue) [1]H and [13]C chemical shifts are shown. Central methines could not be assigned.

The online version of this article includes the following figure supplement(s) for figure 2:

**Figure supplement 1.** Ms-Ms fragmentation spectrum of *m/z* = 323.1647 in extracts of *F. perrara* wt.

**Figure supplement 2.** UV spectrum of the extracts indicates isomerization of the aryl polyene.

**Figure supplement 3.** [1]H nuclear magnetic resonance (NMR) spectrum of enriched aryl polyene in DMSO-$\delta_6$ at 298 K.

**Figure supplement 4.** [1]H nuclear magnetic resonance (NMR) spectrum of enriched aryl polyene in DMSO-$\delta_6$ at 298 K.

**Figure supplement 5.** Heteronuclear Single Quantum Coherence (HSQC) spectrum of enriched aryl polyene in DMSO-$\delta_6$ at 298 K.

**Figure supplement 6.** Heteronuclear Single Quantum Coherence (HSQC) spectrum of enriched aryl polyene in DMSO-$\delta_6$ at 298 K.

**Figure supplement 7.** COrrelated SpectroscopY (COSY) spectrum of enriched aryl polyene in DMSO-$\delta_6$ at 298 K.

**Figure supplement 8.** COrrelated SpectroscopY (COSY) spectrum of enriched aryl polyene in DMSO-$\delta_6$ at 298 K.

**Figure supplement 9.** Heteronuclear multiple-bond coherence (HMBC) spectrum of enriched aryl polyene in DMSO-$\delta_6$ at 298 K.

**Figure supplement 10.** Chemical shifts of enriched aryl polyene in DMSO-$\delta_6$ at 298 K.

**Figure supplement 11.** HPLC-High Resolution electrospray ionisation mass spectrometry (HRESIMS) analysis of *F. perrara* and *E. coli* CFT073.

CFUs than the *ihfA** mutant (*Figure 3C*, Wilcoxon rank-sum test p-value<0.0001). In fact, in 50% of all bees (n = 36) the colonization levels of *ihfA** were below the detection limit of 500 CFUs (*Figure 3C*).

As the quantification of *F. perrara* was based on CFUs obtained for whole-gut tissue, we carried out a second experiment, in which we specifically assessed the colonization levels in the pylorus and the ileum region of the honey bee gut, using both CFUs and quantitative PCR (*Figure 3—figure*

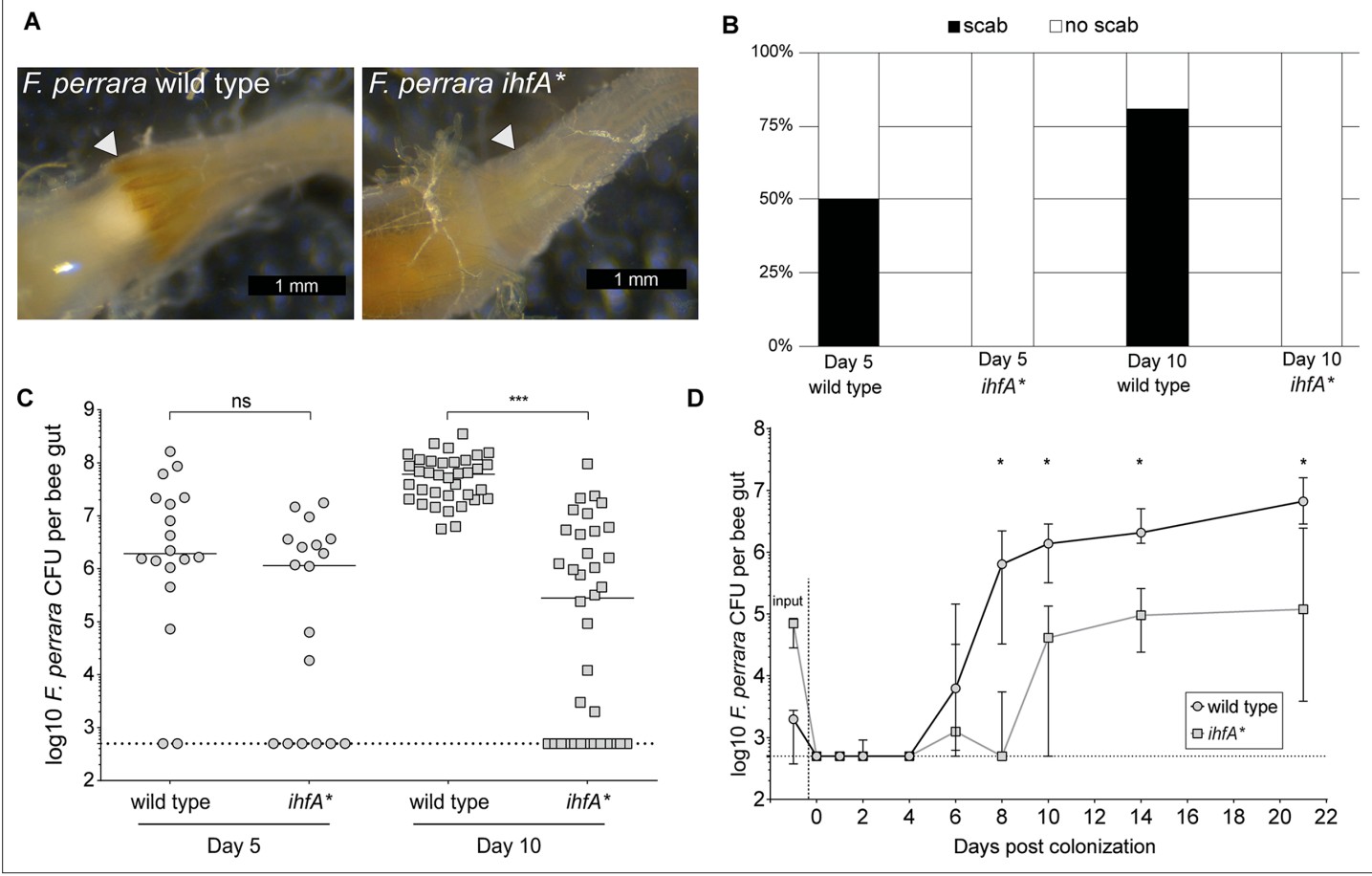

**Figure 3.** *F. perrara ihfA\** mutant displays a colonization defect. (**A**) Light microscopy pictures of pylorus region of bees colonized with *F. perrara* PEB0191wt or *ihf*A\* 10 d post colonization. (**B**) Quantification of scab phenotype of bees 5 and 10 d post colonization with n = 18 and n = 36 per treatment, respectively. (**C**) Quantification of colonization levels is measured by colony-forming units (CFUs) at day 5 (n = 18) and day 10 (n = 36) post colonization. Wilcoxon rank-sum test was used to assess significant differences. (**D**) Time-course experiment of bees colonized with *F. perrara* wt or *ihf*A\*. Colonization levels were measured by CFUs every second day until day 10 and then at day 14 and day 21. n = 12 bees per time point per treatment. Wilcoxon rank-sum test was used to assess significant differences per time point. Error bars represent median and interquartile range. Data from three independent experiments. \*p<0.05, \*\*p<0.01, \*\*\*p<0.001. *Figure 3—source data 1* contains the numeric values for the figures shown here.

The online version of this article includes the following source data and figure supplement(s) for figure 3:

**Source data 1.** Numeric data underlying the results shown in *Figure 3* and *Figure 3—figure supplement 1*.

**Figure supplement 1.** Quantification of *F. perrara* wt and *ihf*A\* in the pylorus and ileum.

*supplement 1*). The results were comparable to those obtained for the whole gut: at day 10 post colonization, there was a significant difference in the colonization levels of the wt and *ihf*A\* in both the pylorus and the ileum (Wilcoxon rank-sum test p-value<0.05).

To obtain a better understanding of the colonization dynamics of *F. perrara* wt and the *ihf*A\* mutant, we conducted a third gnotobiotic bee experiment in which we inoculated microbiota-free bees with one of the two strains and followed the colonization levels over 12 time points from day 0 (i.e., 4 hr post inoculation) until day 22 post inoculation (*Figure 3D*). From the first time point at 4 hr post inoculation until day 4 post inoculation, the bacterial levels were below the detection limit (i.e., below 500 CFUs) in both conditions. Between day 4 and day 8 post inoculation, the abundance of the wt increased rapidly to about $10^6$ CFUs per gut and then steadily further to $10^7$ CFUs per gut until the last time point. In contrast, the levels of the *ihf*A\* mutant remained low until day 10 post colonization and reached on average no more than $10^5$ CFUs per gut until the last time point at day 22 post colonization. Notably, while we had used the same optical density of the two strains for colonizing microbiota-free bees, dilution plating revealed that there were fewer CFUs in the inocula for the wt compared to the *ihf*\* mutant. Despite these differences, the wt colonized much better than *ihf*A\*. In

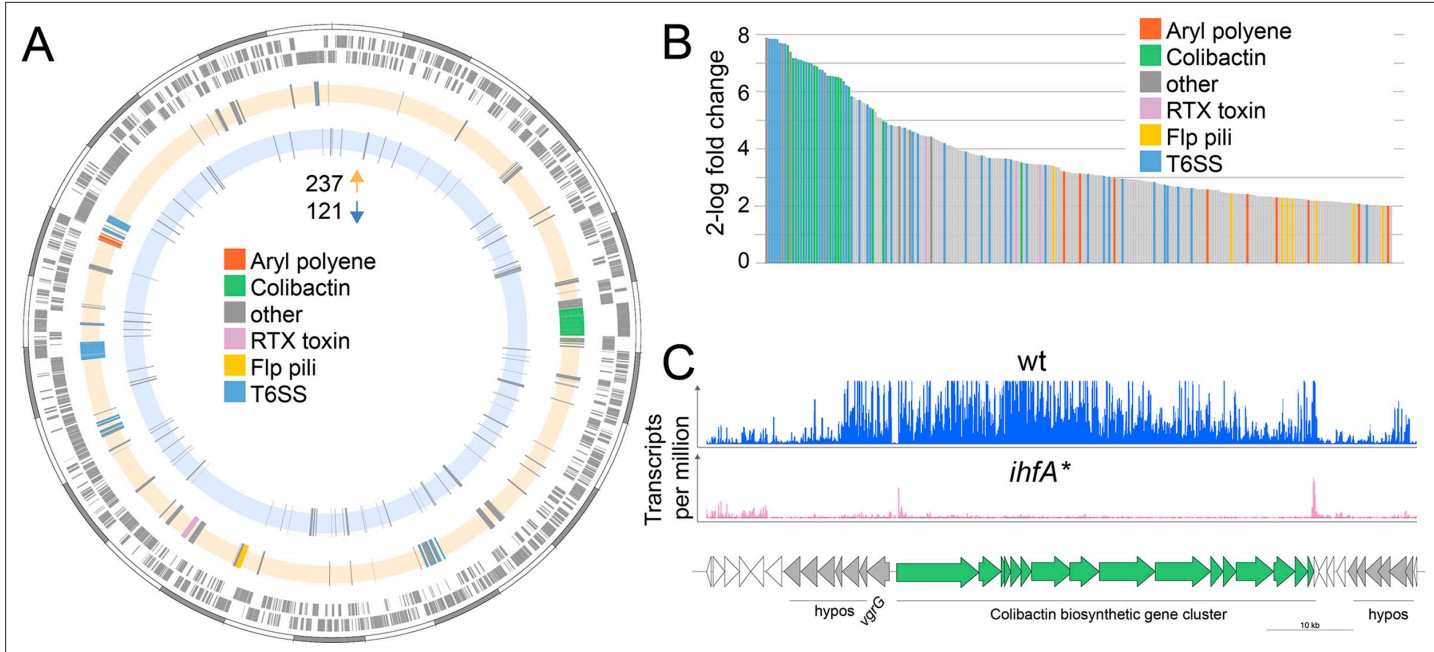

**Figure 4.** Differential gene expression between *F. perrara* wt and *ihfA\** mutant during in vitro growth. (**A**) Chromosomal localization of all genes significantly differentially expressed (2-log fold change = |2|, Fisher's exact test p-value <0.05, false discovery rate [FDR] < 0.05) between *F. perrara* wt and the *ihfA\** mutant. Starting from outside, the first circle shows the scale of the genome representation of *F. perrara* in gray and white steps of 100 kb. The second and third circles (gray) depict the genes on the plus and minus strands of *F. perrara*. The fourth (beige) and fifth (light blue) circle depicts genes upregulated and downregulated in wt compared to *ihfA\**. Genomic islands are highlighted by coloration. (**B**) Bar plot of the genes differentially expressed between *F. perrara* wt and *ihfA\** with a log2-fold change > 2 (Fisher's exact test p-value<0.05, FDR <0.05). (**C**) Comparison of the transcriptional profile of the genomic location encoding the colibactin biosynthetic gene cluster between *F. perrara* wt and the ihfA* mutant. Transcripts per million were visualized using the Integrative Genome Browser (*Freese et al., 2016*). The colibactin operon is schematically depicted below (green arrows). *Figure 4—source data 1* and *Figure 4—source data 2* contain the data used to produce the figure shown here.

The online version of this article includes the following source data for figure 4:

**Source data 1.** List of *F. perrara* genes differentially expressed in vitro.

**Source data 2.** Clusters of orthologous groups for genes differentially expressed in vitro.

summary, these results show that *ihfA\** has a strong colonization defect. It has a delayed colonization dynamics compared to the wt, does not reach the same bacterial loads, and does not cause the scab phenotype, even though the bees were inoculated with more viable cells of *ihfA\** than the wt.

## Genes involved in symbiotic interactions are upregulated in *F. perrara* wt relative to the *ihfA\** mutant

IHF may not have a direct effect on gut colonization, but rather regulate the gene expression of host colonization factors. To test this, we assessed the transcriptional differences by RNA sequencing (RNAseq) between the wt and *ihfA\** mutant when grown in vitro. We found that 358 out of 2337 genes encoded in the genome of *F. perrara* were differentially expressed with a log2-fold change >|2| between the two strains (Fisher's exact test with p<0.05 and false discovery rate (FDR) <5%). Of those, 237 and 121 genes were up- and downregulated in *F. perrara* wt versus *ihfA\**, respectively (*Figure 4A and B*, *Figure 4—source data 1*). Among the genes upregulated in the wt, 'Intra-cellular trafficking, secretion, and vesicular transport' (COG U), 'Extracellular structure' (COG W), 'Lipid transport and metabolism' (COG I), 'Mobilome: prophages and transposases' (COG X), and 'Secondary metabolites biosynthesis, transport and catabolism' (COG Q) were significantly enriched (Fisher's exact test, BH-adjusted p-value<0.01; see *Figure 4—source data 2*). Genes belonging to these three categories include different subunits and effectors of the two T6SSs of *F. perrara*, the Clb biosynthesis gene cluster, various components of the Flp pili, and an RTX (repeats in toxin) toxin belonging to the type I secretion system family (*Figure 4—source data 1*). Also, the genes of the APE biosynthesis gene cluster were among the upregulated genes, which is in line with the

production of the corresponding metabolites in the wt but not in the mutant strain (*Figure 2*). Interestingly, a relatively large proportion of the upregulated genes encoded hypothetical or poorly characterized proteins. In fact, genes without COG annotation were also enriched relative to the entire genome of *F. perrara* (Fisher's exact test, BH-adjusted p-value<0.01; see *Figure 4—source data 2*). Many of the upregulated genes were organized in genomic islands, with the largest one including the biosynthesis gene cluster of Clb and many hypothetical protein-encoding genes (*Figure 4C*). T6SS and Clb biosynthesis genes were among the genes with the highest fold changes relative to *ihfA\** mutant (*Figure 4B*, 28 of 32 genes with log2-fold change > 6). Moreover, 64% of the upregulated genes (152/237) belonged to the *F. perrara*-specific gene content as based on our previously published genome comparison of *F. perrara* PEB0191 with four other strains of the family Orbaceae (three of the genus *Gilliamella* and one of the genus *Orbus*, *Figure 4—source data 1*; *Engel et al., 2015b*).

Among the 121 downregulated genes, only COG category O ('Posttranslational modification, protein turnover, chaperones') was statistically enriched (Fisher's exact test, BH-adjusted p-value<0.01, see *Figure 4—source data 2*). Moreover, only a small fraction (12%) belonged to the '*F. perrara*-specific genes,' and fewer genes were organized into genomic islands. A more detailed inspection of the annotation revealed that a large number of the downregulated genes were involved in transport and metabolism (40 genes), transcriptional regulation (10 genes), and protein folding (8 genes), highlighting clear differences in the functional roles of the up- and downregulated genes. The two genes with the highest fold change (log2-fold change < -5) both encoded transcriptional regulators. One of them, *dksA* (Fpe_01158), is located upstream of the *mrsA/mrsB* antioxidant system (Fpe_01159 to Fpe_01162), which was also among the downregulated genes. The other one is part of the two-component regulator system *basS/basR* (Fpe_02097 and Fpe_02098), which has been reported to act as an iron- and zinc-sensing transcriptional repressor and activator in *E. coli* (*Lee et al., 2005*; *Hantke, 2001*). Taken together, these results show that many accessory genes known to be involved in symbiotic interactions (colibactin, Flp pili, T6SS) are upregulated in *F. perrara* wt as opposed to *ihfA\**, providing a list of candidate genes responsible for the colonization defect of the *ihfA\** mutant.

## T6SS, pili, APE biosynthesis, and Clb biosynthesis genes are expressed during bee gut colonization

To test whether the genes upregulated in vitro in the wt relative to *ihfA\** were expressed in vivo, we determined the transcriptome of *F. perrara* wt at day 5 and day 10 post colonization. A total of 260 (149 up and 111 down) and 298 (162 up and 136 down) genes were differentially expressed at day 5 and day 10 post colonization relative to growth in vitro (log2-fold change >|2|, quasi-likelihood F-test with p<0.05 and FDR < 5%, *Figure 5—source data 1*). There was a considerable overlap of the differentially regulated genes between the two time points (115 and 80 shared up- and downregulated genes, respectively). At both time points, the COG category 'Carbohydrate transport and metabolism' (COG G) was significantly enriched among the genes upregulated in vivo relative to the entire genome (*Figure 5—source data 2*). In addition, at time point day 10, also the COG category (P) 'Inorganic ion transport and metabolism' was enriched (P adj<0.01, Fisher's exact test, *Figure 5—source data 2*). Genes belonging to these two categories encoded transporters for different sugars (Phosphotransferase systems), iron, and transferrin (*Figure 5—figure supplement 1*; *Figure 5—source data 1*). In addition, a catalase gene and several genes for the biosynthesis of the amino acid tryptophan were upregulated at both time points. However, only 14 and 19 genes of those upregulated in vitro in the wt relative to the *ihfA\** mutant (see *Figure 5*) were also upregulated in vivo at day 5 and day 10 post-colonization, respectively (*Figure 5—source data 1*). This was expected because the in vitro RNAseq analysis had shown that these genes are already expressed in the wt when grown on mTYG agar, which we used as a reference condition for the in vivo analysis. Indeed, when comparing count-normalized gene expression (as measured by transcripts per million [TPM]) across the different conditions, we found that most of the T6SS machinery, APE biosynthesis, pilus, and iron uptake genes were expressed at both time points in vivo, and to similar levels as in vitro (*Figure 5*, *Figure 5—figure supplement 2*). Only the Clb genomic island and some of the VgrG-like T6SS effector genes had clearly lower TPM values in vivo than in vitro, yet higher than in *ihfA\** in vitro (*Figure 5C and F*). These results suggest that most of the genes upregulated in vitro in the wt relative to *ihfA\** are also expressed at high level by the wt in vivo.

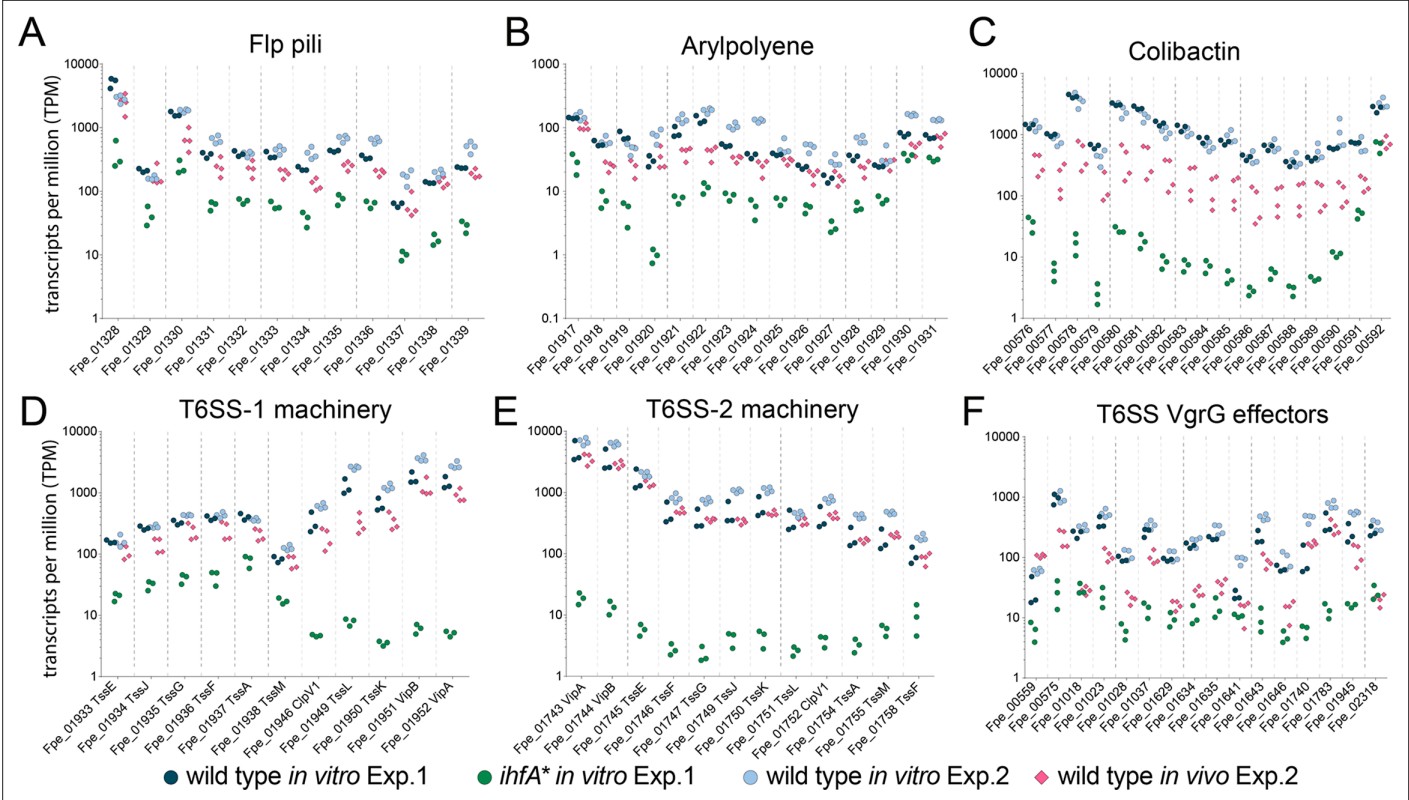

**Figure 5.** Gene expression of Ihf-regulated genes of *F. perrara* 10 d post inoculation of gnotobiotic honey bees. Transcripts per million were calculated for all replicates of the in vitro and the in vivo RNAseq experiments. For the in vitro experiment (Exp. 1), all three replicates of the wt and the *ihfA\** mutant are shown. For the in vivo experiment (Exp. 2), the four replicates of the day 10 time point and the in vitro reference condition are shown. Data for day 5 time point in comparison to day 10 time point is shown in *Figure 5—figure supplement 2*. *Figure 5—source data 1* and *Figure 5—source data 2* contain the data used to generate the figure shown here. (**A-F**) The expression of Flp pili genes (**A**), genes involved in the synthesis of Arylpolyene (**B**) and Colibactin (**C**) and genes involved in the function of T6SSs (**D-F**) is shown.

The online version of this article includes the following source data and figure supplement(s) for figure 5:

**Source data 1.** List of *F. perrara* genes differentially expressed in vivo.

**Source data 2.** Clusters of orthologous groups for genes differentially expressed in vivo.

**Figure supplement 1.** RNAseq comparison of *F. perrara* during in vivo colonization compared to growth in vitro.

**Figure supplement 2.** Gene expression of Ihf-regulated genes of *F. perrara* 5 d post inoculation of gnotobiotic honey bees.

## Gene deletion of IhfA-regulated genes result in impaired gut colonization and/or abolish scab development

To test the direct impact of IhfA-regulated genes on host colonization and/or scab development, we established a gene-deletion strategy for *F. perrara* based on a two-step homologous recombination procedure (see 'Materials and methods' and *Figure 6—figure supplement 1*). This allowed us to create six different non-polar in-frame gene-deletion mutants of potential colonization factors regulated by IHF (Key Resources Table). We deleted an essential gene of the colibactin biosynthesis pathway (Δ*clbB*), both *hcp* genes of the two T6SSs, either separately or as double mutant (Δ*hcp1*, Δ*hcp2*, and Δ*hcp1*/Δ*hcp2*), the gene encoding the major Flp pili subunit (Δ*pilE*), and the entire APE biosynthesis gene cluster (Δ*apeA-R*). Deletion mutants were confirmed by genome resequencing. The Δ*apeA-R* mutant was the only strain not forming yellow colonies anymore (*Figure 1—figure supplement 1*), which is consistent with the idea that the aryl polyene pathway is responsible for the yellow color. We measured the growth of the gene-deletion mutants in vitro, which was similar to the wt and *ihfA\** strains (*Figure 6—figure supplement 2*). Additionally, no significant differences were observed in cell length (*Figure 6—figure supplement 3*). Moreover, we corresponded $OD_{600}$

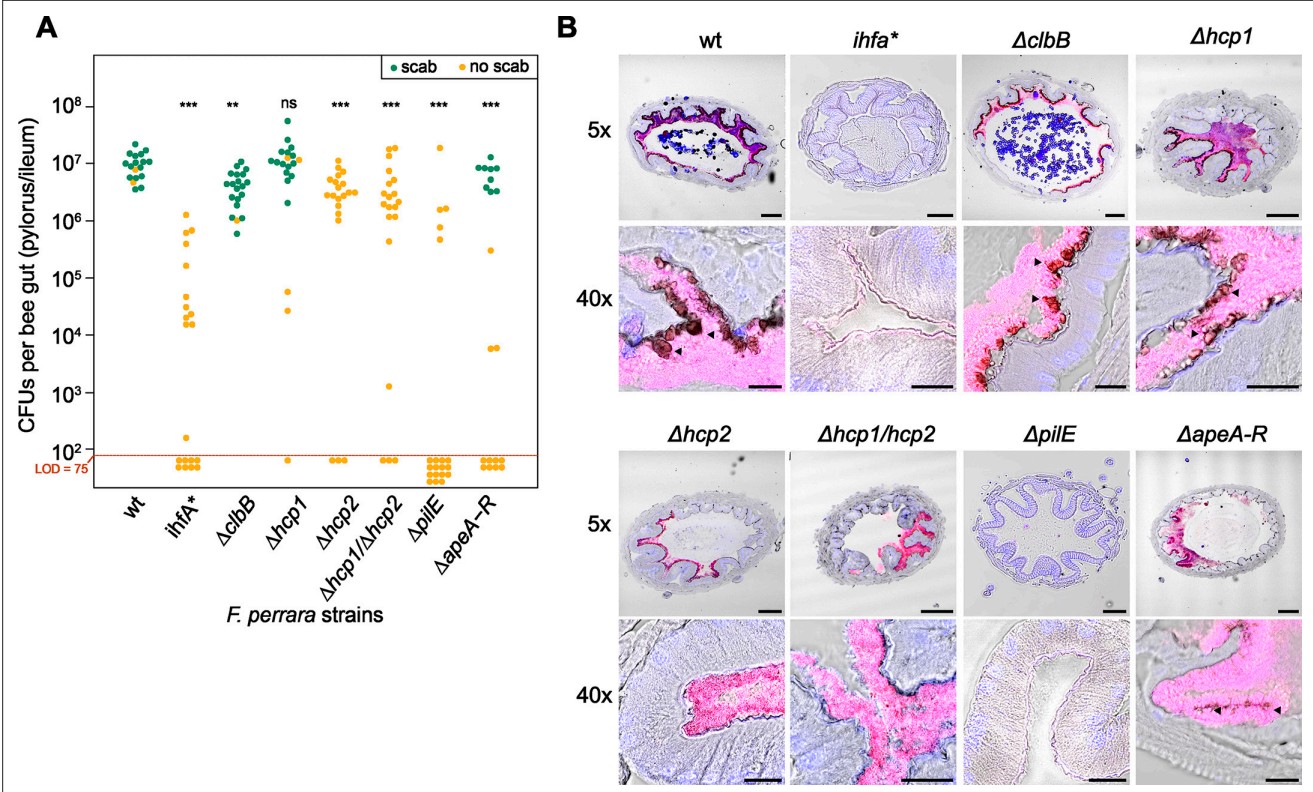

**Figure 6.** Gut colonization phenotypes of different gene-deletion mutants of *F. perrara*. (**A**) Colonization levels were assessed 10 d after inoculation by counting colony-forming units (CFUs) in dilutions of homogenized bee guts plated on Brain Heart Infusion (BHI) agar. Only the pylorus and ileum section of the gut were analyzed. Limit of detection (LOD) corresponds to the lowest colonization level detectable in our assay, that is, points below the LOD correspond to bees for which no CFUs were detected. Statistically significant differences of the colonization levels of each mutant relative to the wt of *F. perrara* were determined using the Wilcoxon rank-sum test with BH correction. Bees were inoculated with an $OD_{600}$ of 0.1. Data come from two independent experiments. *Figure 6—figure supplement 5* shows the data points by experiments. *p<0.05, **p<0.01, ***p<0.001. Filled circle colors indicate whether a scab was detected during dissection (green = scab; yellow = no scab). (**B**) Location within the pylorus was assessed using FISH microscopy. Bees were inoculated with different *F. perrara* genotypes at $OD_{600}$ = 0.1, guts were dissected at day 10 after inoculation and sectioned using a microtome. Hybridizations were done with probes specific for *F. perrara* (magenta). DAPI counterstaining of host nuclei and bacteria is shown in blue. Images were generated by merging brightfield, *F. perrara* and DAPI images that were obtained for the same section of the gut. The composite images here shown were obtained by merging the images of each channel presented in *Figure 6—figure supplements 6 and 7*. These were obtained using the ×5 and ×40 objectives of the Zeiss LSM900. Scale bar for images obtained with ×5: 100 µm, for ×40: 20 µm. *Figure 6—source data 1* contains the numeric values used to generate (**A**).

The online version of this article includes the following source data and figure supplement(s) for figure 6:

**Source data 1.** Numeric data underlying the results shown in *Figure 6* and *Figure 6—figure supplements 3–5*.

**Figure supplement 1.** Scheme of the gene-deletion strategy based on a two-step homologous recombination procedure.

**Figure supplement 2.** Growth curves for the different *Frischella* strains.

**Figure supplement 3.** Single-cell imaging of *F. perrara* strains.

**Figure supplement 4.** Correspondence between OD and colony-forming unit (CFU) for *F. perrara* genotypes.

**Figure supplement 5.** Same graph as shown in *Figure 6* but data points are labeled by experiment.

**Figure supplement 6.** *F. perrara* colonization of the pylorus.

**Figure supplement 7.** *F. perrara* colonization of the pylorus.

to CFU counts and found that both the Δ*hcp1*/Δ*hcp2* and Δ*pilE* strains had lower counts than the wt strain (*Figure 6—figure supplement 4*). To compare the six gene-deletion strains to *F. perrara* wt and the *ihfA** mutant in terms of gut colonization and induction of the scab phenotype, each strain was inoculated into microbiota-free bees and CFUs assessed in the pylorus/ileum region 10 d post inoculation. As expected, the wt successfully colonized the pylorus/ileum region of all analyzed gnotobiotic bees (median = $9.56 * 10^6 \pm 5.06 * 10^6$ CFUs per gut; n = 18 bees, two independent experiments)

and induced the scab phenotype in 16 of 18 bees (*Figure 6A* and *Figure 6—figure supplement 5*). In contrast, the *ihfA\** colonized poorly compared to the wt (median = $1.46 * 10^4 \pm 3.19 * 10^5$ CFUs per gut, n = 20 bees, Wilcoxon rank-sum test p<0.001), and not a single bee developed the scab phenotype. Of the six tested gene-deletion mutants, only the mutant of T6SS-1 (Δ*hcp1*) reached wt colonization levels and also induced the scab phenotype in most of the bees (15 out of 20). The other five mutants all exhibited lower colonization levels than the wt strain. However, the severity of the colonization defect varied between the mutants, and while some of the mutants still caused the scab phenotype, others did not. For example, the Clb biosynthesis gene cluster mutant Δ*clbB* induced the scab phenotype in all but one bee, despite the fact that the colonization levels were about twofold lower than for the wt (median = $4.24 * 10^6 \pm 2.87 * 10^6$ CFUs per gut). The gene-deletion mutant of the T6SS-2 (Δ*hcp2*) and the double mutant of both T6SSs (Δ*hcp1*/Δ*hcp2*) reached similar colonization levels as the Δ*clbB* mutant. Yet, none of the bees colonized with these two mutants developed the scab phenotype not even those that had very high bacterial counts (*Figure 6A*). The deletion mutant of the APE biosynthesis pathway (Δ*apeA-R*) showed again a different colonization phenotype: for some bees no colonization was detected, while others showed similar colonization levels as the wt of *F. perrara*. However, in contrast to the Δ*hcp2* and the Δ*hcp1*/Δ*hcp2* mutants, the Δ*apeA-R* mutant still induced the scab phenotype, but only in the bees that had high bacterial loads. Finally, the Δ*pilE* mutant had the strongest impact on colonization. Only 5 out of 20 bees had detectable levels of *F. perrara* in the gut (limit of detection: 75 CFUs per gut) and none of the bees developed the scab phenotype. Electron microscopy imaging revealed this strain did not have pili, confirming the Δ*pilE* mutation affects the formation of these structures (*Figure 6—figure supplement 3C*).

The Δ*clbB*, Δ*hcp2,* and Δ*hcp1*/Δ*hcp2* mutants all reached similar colonization levels, yet only colonization with Δ*clbB* led to the development of the scab phenotype. These differences in scab formation could be due to an altered colonization pattern of the Δ*hcp2* and Δ*hcp1*/Δ*hcp2* mutants.

To address this hypothesis, we visualized how the gene-deletion mutants are distributed in the pylorus (*Figure 6B*, *Figure 6—figure supplements 6 and 7*). We obtained cross-sections of the pylorus of bees associated with these mutants, stained them with DAPI and an *F. perrara*-specific FISH probe, and imaged these sections using confocal microscopy. As previously reported *Engel et al., 2015a*, *F. perrara* wt was found to colonize the pylorus region, forming a dense biofilm in close proximity to the host and occupying the crypts. A similar colonization pattern was observed for Δ*hcp1*, Δ*hcp2*, Δ*hcp1*/Δ*hcp2*, Δ*clbB,* and Δ*apeA-R*. In contrast, for the *ihfA\** and Δ*pilE* mutants, we did not identify any bacteria in the analyzed gut sections, which is in agreement with the low colonization levels of these mutants detected by CFU plating. Dark spots corresponding to scab material were found along the cuticular lining colonized by bacteria for the Δ*clbB*, Δ*hcp1,* and Δ*apeA-R,* but not for the Δ*hcp2* and Δ*hcp1*/Δ*hcp2* mutants, which matches the visual inspections of the dissected guts for the presence/absence of the scab phenotype across the different strains (*Figure 6B*). Based on these results, we conclude that the inability of the Δ*hcp2* and Δ*hcp1*/Δ*hcp2* mutants to trigger the scab phenotype cannot be explained by an altered localization in the gut. Overall, these results confirm that IHF regulates various host colonization factors that when deleted cause distinctive colonization defects.

## *F. perrara* T6SS-2 and the APE biosynthesis pathway regulate gut colonization in complex bacterial communities

In natural conditions, *F. perrara* shares its niche with other symbionts of the bee gut. As several factors regulated by IHF are known for their role in microbial interactions (*Basler et al., 2013*; *Gallegos-Monterrosa and Coulthurst, 2021*), we also wanted to test the impact of the six gene-deletion mutants on the ability of *F. perrara* to colonize the bee gut in the presence of other community members. We compared gut colonization levels of the gene-deletion mutants, *ihfA\*,* and the wt in the presence and absence of a synthetic community (BeeComm_002) composed of 13 strains representing major core microbiota members of the honey bee gut microbiota (see Key Resources Table). For five of the eight tested strains, we did not detect a significant effect of the BeeComm_002 on the colonization levels (*Figure 7*, *Figure 7—figure supplement 1*). This included *F. perrara* wt, which successfully colonized the gut independently of the presence of the BeeComm_002 (Wilcoxon rank-sum test p=0.899, two independent experiments), the *ihfA\** (Wilcoxon rank-sum test p=0.638), and the pili mutant Δ*pilE* (Wilcoxon rank-sum test p=0.217), which both already exhibited a strong colonization

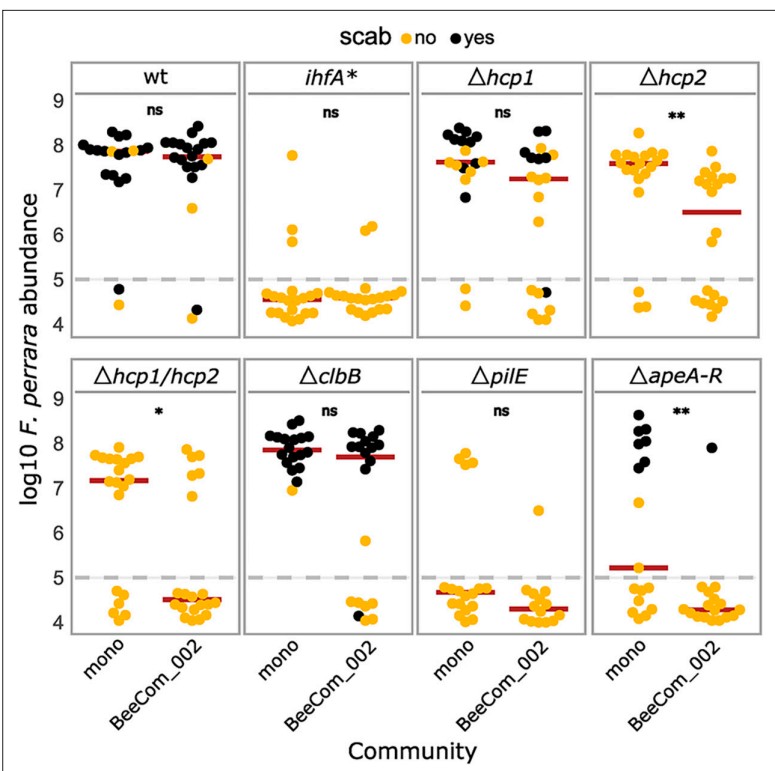

**Figure 7.** Gut colonization of the gene-deletion mutants in the presence of bacterial competition. Bees were inoculated with *F. perrara* alone (mono) or in the presence of a defined bacterial community representing core members of the bee gut microbiota (BeeCom_002). Colonization levels were assessed 10 d after inoculation by qPCR. Only the pylorus and ileum section of the gut were analyzed. The dashed gray line refers to the limit of detection (LOD) and corresponds to the lowest colonization level detectable in our assay, that is, points below the LOD correspond to bees for which no *F. perrara* was detected. Statistically significant differences between the colonization levels of each mutant in mono-association compared to in the presence of the defined microbial community were determined using the Wilcoxon rank-sum test with BH correction. Data comes from two independent experiments. *Figure 7—figure supplement 1A* shows the data points colored by experiments. *p<0.05, **p<0.01, ***p<0.001. Filled circle colors indicate whether a scab was detected during dissection (black = scab; yellow = no scab). *Figure 7—source data 1* contains the numeric values used to generate *Figure 6A*.

The online version of this article includes the following source data and figure supplement(s) for figure 7:

**Source data 1.** Numeric data underlying the results shown in *Figure 7* and *Figure 7—figure supplement 1*.

**Figure supplement 1.** Colonization of wt, ihfa, and gene-deletion mutants of *F. perrara*.

defect in mono-colonization, and the colibactin gene cluster mutant Δ*clbB* (Δ*clbB:* Wilcoxon rank-sum test p=0.127) and the T6SS-1 mutant Δ*hcp1* (Wilcoxon rank-sum test p=0.068), which both colonized at equal levels in both conditions (*Figure 7*). For the other three mutants, we observed a significant reduction in colonization levels in the presence of the BeeComm_002. The colonization levels of T6SS-2 mutant Δ*hcp2* decreased sevenfold (median mono-association: $3.86 * 10^7 ± 4.16 * 10^7$, median BeeComm_002: $5.10 * 10^6 ± 1.77 * 10^7$, Wilcoxon rank-sum test p=0.002), whereas both the T6SS double mutant Δ*hcp1/hcp2* (Wilcoxon rank-sum test p=0.029) and the APE biosynthesis mutant Δ*apeA-R* (Wilcoxon rank-sum test p=0.004) failed to colonize most bees tested (*Figure 7* and *Figure 7—figure supplement 1B*). These results demonstrate that the T6SS-2 and the APE biosynthesis pathway plays a role in regulating *F. perrara* colonization in the presence of other symbionts, possibly regulating the interaction with these bacterial partners.

## Discussion

In this study, we identified genetic factors that allow the gut symbiont *F. perrara* to colonize its specific niche in the pylorus of the honey bee gut and to induce the scab phenotype. Our results advance the

understanding of the genetic factors that facilitate symbionts to colonize niches in the animal gut. Specifically, we find that the DNA-binding protein IhfA plays an important role in gut colonization. IhfA is a histone-like, nucleoid-associated protein (NAP) that forms a heterodimer together with IhfB. IHF binds to and bends DNA in a sequence-specific manner (*Freundlich et al., 1992*; *Goosen and van de Putte, 1995*; *Rice et al., 1996*; *Zulianello et al., 1995*), thereby facilitating the recombination of mobile DNA elements (*Freundlich et al., 1992*; *Goosen and van de Putte, 1995*) and influencing gene expression. We identified three different mutations in IHF, all located in the region of the protein interacting with the DNA. All three IHF mutants formed larger colonies than the wt strain on mTYG agar. Together, this suggests that the identified mutations change the DNA-binding properties of IHF, resulting in broad transcriptional changes in *F. perrara* that provide a growth advantage in vitro relative to the wt strain.

*F. perrara* colonizes the epithelial surface of the pylorus, where it adheres to the cuticular lining and forms a thick biofilm-like layer (*Engel et al., 2015a*). IHF has been shown to bind to extracellular DNA, which in the case of the human pathogen *Haemophilus influenzae* increases biofilm stability (*Jurcisek et al., 2017*). Therefore, it is possible that IHF has a direct impact on gut colonization of *F. perrara* by stabilizing the biofilm formed on the host epithelium. However, based on our gene expression data and the results of the gene-deletion experiments, it seems more likely that IHF influences gut colonization by regulating downstream genes involved in host interaction. Several of the genes regulated by IHF are known to play important roles in the adhesion to surfaces or the formation of biofilms. For example, pili are key factors for adhesion across many bacteria (*Utada et al., 2014*; *Tomich et al., 2007*; *Shime-Hattori et al., 2006*; *Craig et al., 2004*) and have already been shown in the case of the bee gut symbiont *S. alvi* to be beneficial for gut colonization and biofilm formation (*Powell et al., 2016a*). In agreement with these previous results, several of the deletion mutants of IHF-regulated genes showed colonization defects in our gnotobiotic bee experiments corroborating that IHF impacts gut colonization through its effect on gene expression rather than through a direct role in biofilm formation.

The pili mutant (Δ*pilE*) had the strongest colonization defect of the tested gene-deletion strains, likely because pili mediate the adhesion to the host epithelium, and therefore allow *F. perrara* to persist and replicate in the pylorus/ileum region of the honey bee gut. The deletion mutant of the APE biosynthesis pathway (Δ*apeA-R*) also showed a clear colonization defect. However, while some of the bees inoculated with this mutant had no detectable levels of *F. perrara* in the gut, others reached similar levels as bees colonized with the wt strain and also developed the scab phenotype. Aryl polyenes can protect bacteria from reactive oxygen species (*Schöner et al., 2016*; *Johnston et al., 2021*), which are produced by insects as part of the host immune response in the midgut (*Kuraishi et al., 2013*; *Ha et al., 2005*). Our time-course experiment revealed that in the first few days of colonization the bacterial numbers of *F. perrara* in the gut are much lower than in the initial inoculum (*Figure 2*), suggesting that the bacteria experience a significant population bottleneck at the beginning of the colonization process. It is possible that the Δ*apeA-R* mutant is impaired in its ability to resist reactive oxygen species (or other physicochemical stressors) when passing through the anterior sections of the honey bee gut and therefore reaches its niche in the pylorus in only a fraction of the inoculated bees. These stressors may be even more active in the presence of a complex community, which could explain the even lower colonization success of Δ*apeA-R* in the presence of the BeeComm_002. However, APEs have also been implicated in other functions such as biofilm formation (*Johnston et al., 2021*). Therefore, future studies will be needed to elucidate how the APE biosynthetic pathway of *F. perrara* impacts honey bee gut colonization. Given the widespread occurrence of these biosynthetic genes throughout different bacteria, this might help to understand their role in host colonization in a much wider context (*Cimermancic et al., 2014*).

A somewhat unexpected result of our mutant analysis was that the deletion of one of the genes encoding an essential NRPS/PKS enzyme (ClbB) of the Clb biosynthetic gene cluster did not affect scab development and had only a weak impact on gut colonization (twofold lower levels than wt). Studies on *E. coli* have shown that the Clb biosynthesis pathway induces DNA damage in eukaryotic cells (*Nougayrède et al., 2006*) and contributes to tumorigenesis in the mammalian gut (*Cuevas-Ramos et al., 2010*; *Arthur et al., 2012*). We have previously shown that *F. perrara* also induces DNA damage in eukaryotic cells and that this is dependent on a functional Clb biosynthesis pathway (*Engel et al., 2015b*). Therefore, we had speculated that the genotoxic activity of colibactin may trigger the

local melanization response and the development of the scab phenotype upon colonization with *F. perrara* (***Emery et al., 2017***; ***Engel et al., 2015a***). However, the results presented here show that colibactin is not causing the scab phenotype. Therefore, other characteristics of *F. perrara* must explain why this bacterium causes the scab phenotype.

Usually, bees that have <10$^6$ bacteria in the gut do not develop visible scab at day 10 post colonization (***Figures 3 and 7***), which suggests that a certain number of bacteria is needed to elicit this characteristic host response. On the contrary, not all bees with high levels of colonization developed a scab phenotype, indicating that high loads are necessary but not sufficient to trigger the development of a visible scab phenotype. Specifically, the *Δhcp2* mutant of T6SS-2 and the *Δhcp1/Δhcp2* double mutant of T6SS-1 and T6SS-2 both reached relatively high colonization levels, yet not a single bee developed the scab phenotype. T6SS are usually involved in interbacterial warfare by injecting toxins into neighboring bacterial cells (***Mougous et al., 2006***; ***Coyne and Comstock, 2019***; ***Coulthurst, 2019***). However, there is an increasing number of studies showing that certain T6SS effectors can also target eukaryotic cells and modify diverse eukaryotic processes, including adhesion modification, stimulating internalization, cytoskeletal rearrangements, and evasion of host innate immune responses (***Monjarás Feria and Valvano, 2020***). While *F. perrara* is not the only bee gut symbiont harboring T6SS, the effector protein repertoires differ tremendously between different species and even strains of the same species (***Steele and Moran, 2021***). Thus, it is possible that some of the effector proteins of *F. perrara* may target the host rather than other bacteria eliciting the melanization response and scab phenotype in the bee gut. Additionally, both the *Δhcp2* and the *Δhcp* double mutant had increased colonization defects in the presence of the BeeComm_002. This raises the hypothesis that the T6SS-2 may be important for *F. perrara* to interact with other gut symbionts, namely bacteria of the genera *Snodgrassella* and *Gilliamella* that also colonize the pylorus. It is important to mention that our experimental design favored *F. perrara* over the individual members of the BeeComm_002 as, in the inoculum fed to the bees, *F. perrara* was 13 times more abundant than any member of the defined community. It is likely that a stronger colonization defect would be observed if the proportions of *F. perrara* to other community members were more even. In any case, the presence of the BeeComm_002 led to a reduction in the number of bees colonized by most *F. perrara* strains, but not for the wt (***Figure 7—figure supplement 1B***). This is particularly interesting in the case of the *Δhcp1* and *ΔclbB* mutants, suggesting a possible role for these genes in the presence of other symbionts.

We did not carry out a genome-wide screen for host colonization factors and only generated deletion mutants of a few of the IHF-regulated genes. Thus, there are probably many other factors that also contribute to gut colonization. For example, our in vivo RNAseq experiment indicated that metabolic genes involved in tryptophan biosynthesis, sugar transport, and iron uptake are upregulated during gut colonization. This is in line with the TnSeq screen performed in *S. alvi,* which revealed that genes of essential amino acid biosynthesis pathways and iron uptake are important colonization factors of the honey bee gut (***Powell et al., 2016b***). It is, however, remarkable that *F. perrara* only upregulates the tryptophan biosynthesis pathway, which suggest a specific demand for the production of this specific amino acid during gut colonization.

While our study revealed that IHF is important for regulating host colonization factors in the honey bee gut, it remains to be elucidated whether IHF is a direct regulator of the identified genes. In *Vibrio fluvialis*, IHF binding sites were identified upstream of several T6SS gene clusters, indicating that direct transcriptional regulation of similar genes by IHF exists in other bacteria (***Pan et al., 2018***). Alternatively, IHF may act upstream of another regulator that controls the expression of the identified genes. In the plant pathogen *Dickeya zeae*, IHF was suggested to positively regulate different virulence factors through binding to the promoter region of a diguanylate cyclase gene, increasing the production of the secondary messenger c-di-GMP (***Chen et al., 2019***). In *F. perrara*, we found that the transcriptional regulators BasR and DksA were substantially downregulated in the wt versus the *ihfA\** mutant. Hence, it is possible that at least some of the differentially regulated genes may not be under the direct control of IHF but regulated via BasR or DskA.

In conclusion, we identified important gut colonization factors of the bee gut symbiont *F. perrara* and show that they are regulated by IHF. The wide occurrence of these genes in host-associated bacteria suggests similar roles in other environments and calls for a more detailed functional characterization. Our approach to create clean gene-deletion mutants expands the available genetic toolbox

for bee symbionts (*Leonard et al., 2018*) and will help to dissect the molecular mechanisms of the identified gut colonization factors in this animal model.

# Materials and methods

## Key resources table

| Reagent type (species) or resource | Designation | Source or reference | Identifiers | Additional information |
|---|---|---|---|---|
| Strain, strain background (*Frischella perrara*) | *Frischella perrara* wt | **Engel et al., 2013** (10.1099/ijs.0.049569-0) | PEB0191 or ESL0157 | Type strain of *Frischella perrara* isolated from *Apis mellifera* |
| Genetic reagent (*F. perrara*) | *Frischella perrara ihfA** | This study | ESL0158 | Spontaneous P83L mutation (CCA→CTA) in *ihfA* in the wt background; mutated gene: Fpe_00769 |
| Genetic reagent (*F. perrara*) | *Frischella perrara ΔapeO ihfB** | This study | ESL0910 | Spontaneous P82L mutation (CCA→CTA) in *ihfB* in a *ΔapeO* background, mutated genes: Fpe_00778 and Fpe_01928 |
| Genetic reagent (*F. perrara*) | *Frischella perrara ΔapeE ihfA** | This study | ESL0922 | Spontaneous L38S mutation (TTA→TCA) in *ihfA* in a *ΔpilE* background, mutated genes: Fpe_00769 and Fpe_01328 |
| Genetic reagent (*F. perrara*) | *Frischella perrara Δhcp1* | This study | ESL0854 | In-frame deletion of T6SS gene *hcp1* in wt background; locus_tag of gene: Fpe_01742 |
| Genetic reagent (*F. perrara*) | *Frischella perrara Δhcp2* | This study | ESL0855 | In-frame deletion of T6SS gene *hcp2* in wt background; locus_tag of gene: Fpe_01942 |
| Genetic reagent (*F. perrara*) | *Frischella perrara Δhcp1/Δhcp2* | This study | ESL0856 | In-frame deletion of of *hcp1* and *hcp2* of both T6SSs in wt background; locus_tags of genes: Fpe_01742 and Fpe_01942 |
| Genetic reagent (*F. perrara*) | *Frischella perrara ΔclbB* | This study | ESL0888 | In-frame deletion of *clbB* of the colibactin genomic island in wt background; locus_tag of gene: Fpe_00576 |
| Genetic reagent (*F. perrara*) | *Frischella perrara ΔpilE* | This study | ESL0921 | In-frame deletion of *pilE* of the Flp pili in wt background; locus_tag of gene: Fpe_01328 |
| Genetic reagent (*F. perrara*) | *Frischella perrara ΔapeA-R* | This study | ESL0957 | In-frame deletion of *the aryl polyeene biosynthetic gene cluster* in wt background; locus_tags of genes: Fpe_01915–1932 |
| Strain, strain background (*Snodgrassella alvi*) | *Snodgrassella alvi* wkB2 | **Kwong and Moran, 2013** (10.1099/ijs.0.044875-0, 23041637) | wkB2 or ESL0145 | Community member of BeeComm_002; type strain of *Snodgrassella alvi* |
| Strain, strain background (*Gilliamella apicola*) | *Gilliamella apicola* wkB1 | **Kwong and Moran, 2013** (10.1099/ijs.0.044875-0, 23041637) | ESL0309 | Community member of BeeComm_002; *Apis mellifera* isolate of *Gilliamella apicola* |
| Strain, strain background (*Gilliamella apis*) | *Gilliamella apis* ESL0178 | **Ellegaard and Engel, 2018** (10.1128/MRA.00834-18) | ESL0178 | Community member of BeeComm_002; *Apis mellifera* isolate of *Gilliamella apis* |
| Strain, strain background (*Gilliamella sp.*) | *Gilliamella sp.* ESL0177 | **Ellegaard and Engel, 2018** (10.1128/MRA.00834-18) | ESL0177 | Community member of BeeComm_002; *Apis mellifera* isolate of *Gilliamella sp.* |
| Strain, strain background (*Bartonella apis*) | *Bartonella apis* ESL0024 | **Segers et al., 2017** (10.1038/ismej.2016.201) | ESL0024 | Community member of BeeComm_002; *Apis mellifera* isolate of *Bartonella apis* |

*Continued on next page*

*Continued*

| Reagent type (species) or resource | Designation | Source or reference | Identifiers | Additional information |
|---|---|---|---|---|
| Strain, strain background (*Bifidobacterium asteroides*) | *Bifidobacterium asteroides* ESL0197 | **Ellegaard and Engel, 2018** (10.1128/MRA.00834-18) | ESL0197 | Community member of BeeComm_002; *Apis mellifera* isolate of *Bifidobacterium asteroides* |
| Strain, strain background (*B. asteroides*) | *Bifidobacterium asteroides* ESL0198 | (10.1128/MRA.00834-18) | ESL0198 | Community member of BeeComm_002; *Apis mellifera* isolate of *Bifidobacterium asteroides* |
| Strain, strain background (*Lactobacillus* Firm5) | *Lactobacillus* Firm5 ESL0185 | **Ellegaard and Engel, 2018** (10.1128/MRA.00834-18) | ESL0185 | Community member of BeeComm_002; *Apis mellifera* isolate of *Lactobacillus* Firm5 |
| Strain, strain background (*Lactobacillus* Firm5) | *Lactobacillus* Firm5 ESL0183 | **Ellegaard and Engel, 2018** (10.1128/MRA.00834-18) | ESL0183 | Community member of BeeComm_002; *Apis mellifera* isolate of *Lactobacillus* Firm5 |
| Strain, strain background (*Lactobacillus* Firm5) | *Lactobacillus* Firm5 ESL0184 | **Ellegaard and Engel, 2018** (10.1128/MRA.00834-18) | ESL0184 | Community member of BeeComm_002; *Apis mellifera* isolate of *Lactobacillus* Firm5 |
| Strain, strain background (*Lactobacillus* Firm5) | *Lactobacillus* Firm5 ESL0186 | **Ellegaard and Engel, 2018** (10.1128/MRA.00834-18) | ESL0186 | Community member of BeeComm_002; *Apis mellifera* isolate of *Lactobacillus* Firm5 |
| Strain, strain background (*Commensalibacter* sp.) | *Commensalibacter sp.* ESL0284 | **Ellegaard and Engel, 2018** (10.1128/MRA.00834-18) | ESL0284 | Community member of BeeComm_002; *Apis mellifera* isolate of *Commensalibacter* sp. |
| Sequence-based reagent | prPEN0013 | **Zufelato et al., 2004** (10.1016/j.ibmb.2004.08.005) | Forward primer for Actin | TGCCAACACTGTCCTTTCTG |
| Sequence-based reagent | prPEN0014 | **Zufelato et al., 2004** (10.1016/j.ibmb.2004.08.005) | Forward primer for Actin | AGAATTGACCCACCAATCCA |
| Sequence-based reagent | prLK-Frisch-042-F | **Kešnerová et al., 2017** (10.1371/journal.pbio.2003467) | *F. perrara* 16S rRNA gene | GGAAGTTATGTGTGGGATAAGC |
| Sequence-based reagent | prLK-Frisch-043-R | **Kešnerová et al., 2017** (10.1371/journal.pbio.2003467) | *F. perrara* 16S rRNA gene | CTATTCTCAGGTTGAGCCCG |

## Bacterial cultivation

*F. perrara* strains were cultivated on modified tryptone yeast extract glucose (mTYG) medium (0.2% Bacto tryptone, 0.1% Bacto yeast extract, 2.2 mM D-glucose, 3.2 mM L-cysteine, 2.9 mM cellobiose, 5.8 mM vitamin K, 1.4 µM $FeSO_4$, 72.1 µM $CaCl_2$, 0.08 mM $MgSO_4$, 4.8 mM $NaHCO_3$, 1.36 mM NaCl, 1.8 µM hematine in 0.2 mM histidine, 1.25% Agar adjusted to pH 7.2 with potassium phosphate buffer), Columbia Blood Agar (CBA) containing 5% defibrinated sheep blood (Oxoid) or Brain Heart Infusion broth (BHI) and incubated in anaerobic conditions at 34–35°C (8% $H_2$, 20% $N_2$, 78% $CO_2$ in a Vinyl Anaerobic Chamber, Coy Lab). Fresh bacterial cultures were used for each experiment. To this end, pre-cultures streaked out from glycerol stocks were grown for 48 hr and re-inoculated onto fresh

plates for 16–24 hr of growth. For liquid cultures, pre-cultures streaked out from glycerol stocks were grown on TYG or BHIA plates for 48 hr and then inoculated into fresh liquid TYG or BHI. Cultures were incubated at 34–35°C in a ThermoMixer C (Eppendorf) at 800 rpm for 16–24 hr of growth. Strains of *F. perrara* used in this study are listed in Key Resources Table.

## Rearing and experimental colonization of honey bees

Microbiota-depleted bees were generated as described by *Kešnerová et al., 2017*. For experimental mono-colonization, bees were starved for 1–3 hr by removal of the sugar water solution. Then, bees were cooled down to 4°C in a refrigerator or on ice to transfer them (head side first) into 1.5 ml microfuge tubes with a hole at the bottom. Tubes with bees were kept at room temperature (RT). For inoculation, each bee was fed 5 µl of *F. perrara* resuspended in sugar water:PBS (1:1 v/v) through the hole at the bottom of the microcentrifuge tube. The bacterial inocula were adjusted to an $OD_{600}$ of 0.01 or 0.1 depending on the experiment. Colonized bees were kept at 30°C with 70% humidity while having access to sugar water and sterilized bee pollen ad libitum until sampling.

## Tissue dissection and bacterial quantification from colonized honey bees

Bees were anesthetized by exposure to $CO_2$. The whole gut or desired gut tissue, for example, pylorus or ileum was dissected using a scalpel. Malpighian tubules were removed from the gut tissue and the presence of a scab was documented using a dissection stereomicroscope (Leica) as described in *Engel et al., 2015a*; *Emery et al., 2017*. The tissues were placed into 2 ml screw-cap tubes containing glass beads (0.75–1 mm diameter, Roth) and 500–1000 µl PBS depending on the experiment. Homogenization of the sample was done by bead beating (FastPrep-24 5g MP Biomedicals) for 40 s at a speed of 7.5 m/s. Serial dilutions (1:10) were performed for each homogenate and plated onto mTYG or BHI agar. Single colonies were counted to determine the total number of bacteria per gut tissue by multiplying with the dilution factor.

## RNA sequencing of bacterial in vitro cultures

Fresh *F. perrara* cultures were prepared on mTYG plates, harvested, and directly transferred into a tube containing TRI reagent (Sigma-Aldrich, Merck) and silica beads (0.1 mm diameter, Roth). Samples were immediately snap frozen in liquid nitrogen and stored at –80°C until RNA extraction. RNA was extracted using a modified TRI reagent protocol. Samples were homogenized by bead beating with a FastPrep instrument with CoolPrep adapter (FastPrep-24 5G, cooled with dry ice) for two cycles of 45 s at speed of 6 m/s, including a 30 s break between each cycle. Samples were kept at RT for 5 min, subsequently extracted using chloroform, and RNA was precipitated using isopropanol overnight at –20°C. Precipitated RNA was dissolved in RNase-free water (Gibco) and incubated with DNase (NEB) to degrade remaining DNA. RNA samples were cleaned up using NucleoSpin RNA clean up kit (Machery-Nagel) according to the manufacturer's protocol, eluted in RNAse-free water, and stored at –80°C until further use. RNA quality was assessed using Nanodrop, Qubit RNA HS kit (Thermo Fisher Scientific), and Bioanalyzer instrument (Agilent). High-quality RNA samples were sent to Lausanne Genomic Technology Facility (GTF, University of Lausanne) for RNA sequencing. Samples were depleted of 16S rRNA using RiboZero reagent (Illumina) and Truseq stranded-RNA Zero libraries were generated (Illumina) before sequencing on an Illumina HiSeq2500 generating single-end 100 bp reads.

## RNA sequencing of bacteria during honey bee gut colonization

Microbiota-depleted bees were colonized with freshly grown *F. perrara*. A fraction of the inoculum was directly transferred into 2 ml tube containing TRI reagent (Sigma-Aldrich, Merck) and silica beads (0.1 mm diameter, Roth). These samples were immediately snap-frozen in liquid nitrogen and stored at –80°C until RNA extraction, which was carried out in the same way as described in the previous section. For the in vivo RNA samples, bees were colonized with *F. perrara* as described above and sampled at day 5 and day 10 post inoculation. Guts were removed from anesthetized bees, the pylorus and first part of the ileum was dissected, and the presence of the scab was recorded. For each sample, 10 pylorus and ileum sections from bees coming from the same cage were pooled in a 2 ml screw cap tube containing 750 µl of TRI reagent (Sigma-Aldrich, Merck), glass beads (0.75–1 mm

diameter, Roth), and silica beads (0.1 mm diameter, Roth). Immediately after collection the samples were snap-frozen in liquid nitrogen and stored at –80°C until RNA extraction. RNA was extracted as described in the previous section. At each time point, four biological replicate samples were used for RNA sequencing, that is, bees for one sample came from independent cages of gnotobiotic bees. The GTF at the University of Lausanne generated the libraries for the sequencing of the in vivo RNA samples. Poly-A depletion was performed to enrich for bacterial mRNA, and Ribo-zero rRNA depletion was performed to remove prokaryotic and eukaryotic rRNAs. Then TruSeq stranded mRNA libraries (Illumina) were generated. The eight libraries were sequenced on an Illumina HiSeq 2500 to obtain single-end 125 bp reads. Libraries of the in vitro RNA samples were generated as described before. The four libraries corresponding to the *F. perrara* inocula used for colonizing the bees of the four replicates were sequenced on an Illumina MiniSeq at the Department of Fundamental Microbiology (125 bp single-end reads). Two in vivo samples (sample identifier D10_3M and D5_1M) sequenced on the HiSeq 2500 at the GTF were included in the MiniSeq run which confirmed that the two runs gave comparable results.

## Differential gene expression and gene enrichment analysis

Raw FASTQ files provided by the GTF containing all reads and corresponding tags indicating whether they were accepted or filtered out according to the CASAVA 1.82 pipeline (Illumina). Only reads tagged as accepted were kept for further analysis. FASTQC (http://www.bioinformatics.babraham.ac.uk/projects/fastqc/) was used to control the quality of the data, followed by Trimmomatic (Trimmomatic-0.35) to trim adapters. Filtered and trimmed reads were mapped onto the *F. perrara* genome using Bowtie (bowtie2-2.3.2). Mapped reads were quantified using HTseq (version 0.7.2). Differential gene expression analysis was done with the Bioconductor package EdgeR (*Nikolayeva and Robinson, 2014*) using R scripts. For the in vitro RNAseq analysis, negative binomial models were fitted to the data and quantile-adjusted conditional maximum likelihood (qCML) common and tagwise dispersion were estimated. The conditional distribution for the sum of counts in a group was used to calculate the significantly differentially expressed genes using an exact test with an FDR < 5%. Only genes detected in all samples with at least one count per million were used for the analysis. For the in vivo RNAseq analysis, we followed the recommendations specified in the EdgeR user guide for generalized linear models. In short, read counts were normalized by trimmed mean of M-values (*Robinson and Oshlack, 2010*) producing scaling factors used by EdgeR to determine effective library sizes. Then, negative binomial generalized linear models were fitted for each condition and quasi-likelihood F-tests for each defined contrast (i.e., pairwise comparison between conditions) was used to assess the significance of differentially expressed genes. As for the in vitro RNAseq analysis, only genes with mapped reads with at least one count per million in all replicates and conditions were used for the analysis. Genes were considered as differentially expressed upon fulfilling the following criteria: (i) log2 fold change $\geq$ 2; (ii) a p-value of <0.05; and (iii) an FDR < 5%. TPM were calculated as follows *Wagner et al., 2012*:

$$TPM\ of\ a\ gene = \frac{number\ of\ mapped\ reads * read\ length * 10^6}{total\ number\ of\ transcripts\ sampled * gene\ length\ in\ bp}$$

Coverage plots in *Figures 2A and 4C* were generated with bam2wig.pl contained in the Bio-ToolBox-1.68 (https://github.com/tjparnell/biotoolbox, *Parnell, 2021*) and visualized with the Integrated Genome Browser (*Freese et al., 2016*). Scripts for analyzing the RNAseq data can be found under the following Switchdrive link: https://drive.switch.ch/index.php/s/kCNTp4g7n60ffMi.

## DNA extraction of dissected gut tissue homogenates

DNA/RNA extraction was performed using an adapted hot phenol protocol from consisting of sample thawing on ice, homogenization by bead beating at 7.5 m/s for 40 s (FastPrep-24 5g MP Biomedicals), two phenol extractions, and an ethanol precipitation overnight at –80°C. DNA/RNA mixture was pelleted and eluted in RNase-free water (Gibco). Samples were treated with RNaseA, and DNA was purified using a Gel and PCR purification kit (Machery-Nagel). For those samples displayed in *Figure 7* and *Figure 7—figure supplement 1*, samples were homogenized with 165 µl of a solution containing GI lysis buffer, QIAGEN, and lysozyme (10:1 concentration) zirconia beads (0.1 mm dia. Zirconia/Silica beads; Carl Roth) and glass beads in a Fast- Prep24 5G homogenizer (MP Biomedicals) at 6 m/s for 45 s. After homogenization, samples were incubated at 37°C for 30 min. Then, 30 µl of Proteinase

K were added and samples were incubated at 56°C for 1 hr. The purification of nucleic acids was performed using CleanNGS magnetic beads (CNGS-0005) and the Opentron OT-2 pipetting robot. Purified DNA extracts were stored at –20°C until further use.

## Bacterial quantification by qPCR

Bacterial absolute abundances were determined using quantitative PCR (qPCR) assays targeting the 16S rRNA gene of *F. perrara* (**Kešnerová et al., 2017**; **Kešnerová et al., 2020**). Normalization was based on the number of host actin gene copies as described in **Kešnerová et al., 2020**. Primer sequences and other primer characteristics are given in Key Resources Table. qPCR was conducted on a StepOne Plus instrument (Applied Biosystems) with the following run method: a holding stage consisting of 2 min at 50°C followed by 2 min at 95°C, 40 cycles of 15 s at 95°C, and 1 min at 65°C. A melting curve was generated after each run (15 s at 95°C, 20 s at 60°C and increments of 0.3°C until reaching 95°C for 15 s) and used to assess specificity of PCR products. qPCR reactions were performed in 10 µl reactions in triplicates in 96-well plates, and each reaction consisted of 1 µl of DNA, 0.2 µM of forward and reverse primer and 1x SYBR green 'Select' master mix (Applied Biosystems). The qPCR reactions for the data corresponding to *Figure 7* and *Figure 7—figure supplement 1* were carried out in 384-well plates on a QuantStudio 5 (Applied Biosystems). The thermal cycling conditions were as follows: denaturation stage at 50°C for 2 min followed by 95°C for 2 min, 40 amplification cycles at 95°C for 15 s, and 60°C for 1 min. Each reaction was performed in triplicate in a total volume of 10 µl (0.4 µM of each forward and reverse primer; 5 µM 1x SYBR Select Master Mix, Applied Biosystems; 1 µl DNA). Each DNA sample was screened with two different sets of primers targeting either the actin gene of *A. mellifera*, or the universal 16S rRNA region.

For each target, standard curves were generated for absolute quantification using serial dilutions (from $10^7$ to 10 copies) of the target amplicon cloned into the plasmid vector. Absolute abundance of bacteria was calculated using the standard curve and was normalized by the median actin copy number per condition (to account for differences in gut size) and by the amount of 16S rRNA copies per genome. For the calculation, we used the following formula:

$$\textit{Absolute bacterial abundance} = 10^{\left(\frac{Ct_{target} - Intercept}{Slope}\right)} * \textit{dilution factor} \div \textit{number of } 16S \textit{ rRNA copies} \div \left(\textit{median of actin copies per condition}\right)$$

where $Ct_{target}$ is the cycle threshold of the target bacterium, and intercept and slope correspond to the values calculated for the standard curve of the target bacterium.

## Single-cell microscopy

*F. perrara* was freshly grown from stock on plate. Subsequently, liquid overnight cultures starting at an $OD_{600}$ 0.1 were inoculated in TYG for the experiment of *Figure 1—figure supplement 2* and in BHI for the experiment of the *Figure 6—figure supplement 3*. Liquid cultures were grown overnight 16–24 hr with shaking at 34–35°C in a ThermoMixer C (Eppendorf) in anaerobic atmosphere. For *Figure 1—figure supplement 2*, $OD_{600}$ was measured and adjusted to 0.1–0.01 and cells were distributed onto small agar patches on a microscopy slide (Milan S.A., Menzel-Gläser). Cells were observed under 1000 times magnification using a ZEISS imager M1 microscope with phase contrast (PH3) condenser. Pictures were acquired randomly on the slides with VisiView software, 8-bit images were corrected with ImageJ, and the contrast was set at min-value of 0 and max-value of 4000. A 10 µM scale bar was added with ImageJ (**Schneider et al., 2012**). For the experiment displayed in *Figure 6—figure supplement 3*, the $OD_{600}$ was adjusted to 0.1 and 10 µl of bacterial solution were distributed onto small patches containing BHI and 1.5% UltraPure Agarose from Invitrogen. Images were obtained using a Nikon ECLIPSE Ti Series inverted microscope coupled with a Hamamatsu C11440 22CU camera and a Nikon CFI Plan Apo Lambda ×100 oil objective (×1000 final magnification). A 20 µM scale bar was added with ImageJ. For both

experiments, the MicrobeJ plugin for ImageJ was used to measure cell size and area using the acquired pictures (*Ducret et al., 2016*).

## Isolation and identification of aryl polyene compounds

*F. perrara* was streaked from the –80°C glycerol stock onto mTYG plates. Plates were placed in an anaerobic tank with a pack of AnaeroGel 3.5L (AN0035A Thermo Scientific) at 37°C and incubated. Under aerobic sterile conditions, colonies were picked and 5 ml of mTYG medium were inoculated in a Hungate tube. After purging the liquid for 1 hr with gas (83% $N_2$, 10% $CO_2$, 7% $H_2$ v/v), bacteria were grown at 37°C. After 1 d of growth, *F. perrara* wt and the *ihfA\** mutant were individually cultivated in Hungate tubes containing 30 ml mTYG for 3 d. Pellets were harvested, extracted with dichloromethane and methanol, then extracts were analyzed by HPLC-HESI-HRMS. The data revealed a strongly UV-Vis absorbent ion peak at *m/z* 323.1647 [M+H]$^+$. To enrich this compound, 400 ml of mTYG were inoculated with 1 ml of pre-culture in a Hungate tube and purged as described above. After 5–10 d, all but a few ml of medium were harvested and 400 ml of fresh mTYG was added under sterile aerobic conditions, purged with anaerobic gas, and incubated at 37°C. A total of 3 l of mTYG were used to obtain 6.53 g of cells, which were stored at –80°C. Light exposure was minimized during all following steps of the aryl polyene enrichment procedure. A mixture of 340 ml dichloromethane and 170 ml of methanol was added to *F. perrara* cell pellets. After 2 hr stirring at RT, the suspension was filtered and 250 ml KOH (0.5 M) were added. After an additional 1 hr of stirring at RT, the pH was set to 6 using $H_2SO_4$ (1 M). The organic layer was washed twice with 500 ml water, once with 250 ml brine, dried over $Na_2SO_4$, and concentrated. The extract was separated by reverse-phase high-performance liquid chromatography (RP-HPLC, Phenomenex Luna 5 µm C18, φ 21.2 × 250 mm, 15.0 ml/min, $\lambda$ = 420 nm) with water +0.1% formic acid (solvent A) and MeCN + 0.1% formic acid (solvent B). Solvent compositions of 5% B for 4 min, a gradient to 95% B for 18 min, 95% B for 9 min, and 5% B for 4 min were used. Fractions around 24 min exhibited strong UV absorption at 420 nm and were further purified by RP-HPLC (Phenomenex Luna 5µ Phenyl-Hexyl, φ 10 × 250 mm, 2.0 ml/min, $\lambda$ = 420 nm) with 80% B for 5 min and a gradient of 80% B to 100% B for 30 min. Peaks around 17 min were combined, concentrated, and analyzed by NMR spectroscopy. HPLC-HESI-HRMS was performed on a Thermo Scientific Q Exactive mass spectrometer coupled to a Dionex Ultimate 3000 UPLC system. NMR spectra were recorded on a Bruker Avance III spectrometer equipped with a cold probe at 500 MHz and 600 MHz for $^1$H NMR and 125 MHz and 150 MHz for $^{13}$C NMR at 298 K. Chemical shifts were referenced to the solvent peaks of DMSO-$\delta_6$ at $\delta_H$ 2.50 ppm and $\delta_C$ 39.51 ppm.

## Structural characterization of the aryl polyene

$^1$H NMR in conjunction with HSQC data suggested approximately 15 methines and aromatic protons, one methoxy group and one methyl group (*Figure 2—figure supplements 3–7*). From the COSY spectrum, four methines of the conjugated double bond system and two of the aromatic protons could be connected (*Figure 2—figure supplements 7 and 8*). HMBC correlations from the methyl group to an aromatic carbon at $\delta_C$ 156 ppm placed it in *ortho*-position of a hydroxy group and next to a singlet aromatic proton. The singlet of the methoxy group was connected to a carbonyl group by HMBC correlations (*Figure 2—figure supplement 9*).

## Construction of gene-deletion mutants

Targeted in-frame gene deletions of *F. perrara* were constructed using a two-step homologous recombination procedure (see *Figure 6—figure supplement 1*). In a first step, suicide plasmids were constructed harboring two adjacent 700–800-bp-long homology regions matching the up- and downstream region of the gene or gene cluster to be deleted. To this end, the two homology regions were amplified using 2 Phanta Max Master Mix (Vazyme) and cloned into pKS2 using Golden Gate assembly. pKS2 is a derivative of pSEVA312 (*Martínez-García et al., 2020*) that contains an sfGFP expression cassette flanked by two BsaI restriction sites in the multiple cloning site. All clonings were done in

*E. coli* DH5a/ λ pir. To validate successful cloning of the two homology regions, PCR and Sanger sequencing was performed. The plasmids derived from pKS2 containing the homology regions were shuttled into the diaminopimelic acid (DAP)-auxotroph *E. coli* JKE201 and delivered to *F. perrara* wt via conjugation using bi-parental mating. To this end, *E. coli* JKE201 carrying the plasmid of interest was cultivated for 16–20 hr at 37°C in LB with 500 µM DAP (Sigma, LB-DAP) and 30 µg/ml chloramphenicol. Subcultures were made by diluting the culture at a ratio of 1:20 into fresh medium and incubated until exponential growth was reached. A fresh culture of *F. perrara* was prepared by growing it for 36 hr on mTYG agar. *F. perrara* and *E. coli* were harvested in PBS and the $OD_{600}$ adjusted to 10. Equal quantities of both bacteria were mixed and spotted onto a cellulose acetate membrane filter (0.2 µm, 25 mm, Huberlab, Sartorius) placed on BHI agar with 500 µM DAP, followed by incubation at 35°C for 6 hr under microaerophilic condition. Filters with bacterial mixtures were removed from the plates with a forceps and placed into a tube with PBS. Bacteria were removed from the filter by pipetting, vortexing, and agitation for 10 min at 1500 rpm (ThermoMixer C, Eppendorf). The bacterial suspension was centrifuged at $8000 \times g$ for 10 min, the supernatant discarded to remove dead bacteria, and the bacterial pellet was washed once with 1000 µl PBS before resuspending in a reduced volume, that is, 100 µl PBS. The bacterial mixture was plated onto CBA and mTYG with selection antibiotic and incubated at 35°C for 5 d under anaerobic condition. Colonies were picked after 4–5 d of incubation and expanded to fresh plates. DNA was isolated and five different PCRs performed to check for successful integration of the plasmid into the chromosomal region of *F. perrara* targeted for deletion. Positive clones (loop-in strains) were stocked in liquid mTYG containing 20% (v/v) glycerol at –80°C until further usage. To select for bacteria that have lost the integrated plasmid, *F. perrara* loop-in strains were grown on mTYG agar and subsequently transformed with plasmid pYE1. pYE1 is a derivative of pBZ485 (**Harms et al., 2017**) containing the restriction enzyme *I-SceI* under the control of the IPTG-inducible *lac* promoter. The activity of the restriction enzyme *I-SceI* will negatively select bacteria containing pKS2-derivatives as two *I-SceI* sites are flanking the multiple cloning sites on pKS2. To generate electrocompetent *F. perrara* cells, the bacteria were cultivated from glycerol stocks on mTYG agar and incubated at 35°C for 2 d under anaerobic condition. Bacteria from plate were inoculated in BHI broth with a starting $OD_{600}$ of 0.1 and grown anaerobically at 40°C for 12–16 hr. Subsequently, growth was stopped by placing cultures onto ice for 15 min. Cells were made electrocompetent by washing twice with ice-cold MOPS solution supplemented with 20% glycerol, with decreasing volumes per wash. Electrocompetent cells were mixed with >500 ng plasmid DNA and incubated for 15 min on ice before transferring into an electroporation cuvette. Electroporation was performed applying a voltage of 2.5 kV with resistance of 200 Ω and capacitance of 25 µF. Bacteria were immediately resuspended in BHI and incubated anaerobically for 6 hr at 35°C to allow phenotypic expression. Bacteria were plated on CBA, mTYG, and/or BHIA plates with 25 µg/ml kanamycin and 100 µM IPTG to select for pYE1 and induce the restriction enzyme *I-SceI*. Plates were incubated in anaerobic atmosphere at 35°C. After 3–5 d of incubation, bacterial colonies were replica plated onto fresh plates containing 15 µg/ml chloramphenicol (selection marker of pKS2-derivatives) or 25 µg/ml kanamycin (selection marker of pYE1) to identify clones that cannot grow on chloramphenicol anymore as a consequence of the loss of the integrated pKS2-derivative. DNA was extracted and a PCR screen performed with primers amplifying the gene targeted for deletion as well as the overspanning region of the deletion. All clones possessing the gene deletion were subsequently expanded on mTYG agar and stocked in glycerol as described above.

## Correspondence between OD and CFU

For the wt, *ihfA\** and the six gene-deletion mutants, bacteria were grown in BHIA plates for 3 d at 35°C in anaerobic conditions. On the day of the experiment, bacteria were harvested and a bacterial solution at $OD_{600} = 0.1$ was prepared per each genotype. Serial dilutions were performed in a 96-well plate by transferring 10 µl of $OD_{600} = 0.1$ into 90 µl of BHI medium followed by several 1:10 dilutions in BHI. For each genotype, 5 µl of all dilutions were plated on BHIA plates, incubated for 3 d at 35°C in anaerobic conditions and quantified to calculate the number of CFUs present in 5 µl of the initial $OD_{600}$

= 0.1 solution. The CFU values present in 5 µl of bacterial solution at $OD_{600}$ = 0.1 are of relevance as they correspond to the volume fed to bees in the colonization experiments. Three experimental replicates were performed.

## Growth curves in liquid cultures

*F. perrara* was freshly grown from stock on BHIA plates. Subsequently, liquid overnight cultures starting at an $OD_{600}$ = 0.1 were inoculated in BHI and grown for 16–24 hr anaerobically with shaking at 35°C and 600 rpm in a ThermoMixer C (Eppendorf) in anaerobic atmosphere. To obtain the growth curves, one 96-well plate was inoculated with different *F. perrara* strains. Each well contained 200 µl of a bacterial solution at $OD_{600}$ = 0.05. Per strain, four wells were inoculated. Absorbance values were measured at 600 nm every 20 min for 72 hr using a BioTek Epoch2 microplate reader. The plate was incubated in anaerobic conditions at 35°C and continuous orbital shaking for the duration of the experiment.

## FISH microscopy

Tissue sections and FISH experiments were performed as previously described *Engel et al., 2015a*. Briefly, the pylorus and ileum of gnotobiotic bees were dissected and fixed for 5 d in Carnoy's solution (ethanol-chloroform-acetic acid, 6:3:1 [vol/vol]). Fixed tissue samples were washed three times for 1 hr in absolute ethanol, then incubated three times for 20 min in xylene, and finally infiltrated with paraffin three times for 1 hr at 60°C. Samples were placed into molds containing melted paraffin and then hardened by placing them into an ice slurry. Paraffin-embedded tissues were cut into serial 5 µm sections with a microtome (Leica), placed on coated microscopy slides, and cleared from paraffin with xylene. Sections were then hybridized overnight with fluorochrome-labeled oligonucleotide probes targeting the 16S rRNA of *F. perrara* and DAPI. Samples were imaged using the Zeiss LSM900 confocal microscope. Per bacterial genotype, one gut was processed and imaged. The *F. perrara* probe used is named PE1_TYE563_G2, has the sequence CCGCTCCAGCTCGCACCTTCGCT, and a Cy3 flurophore.

## Electron microscopy

The wt, *ihfa\**, and *ΔpilE* strains were grown in BHIA plates for 3 d at 34°C in anaerobic conditions. The day before image acquisition, bacteria were harvested from the plates, transferred to liquid BHI at an initial $OD_{600}$ = 0.1, and were grown for 16 hr in anaerobic conditions with shaking. On the day images were acquired, bacteria were diluted 50 times. Each bacteria suspension was adsorbed on a glow-discharged copper 400 mesh grid coated with carbon (EMS, Hatfield, PA) during 1 min at RT. Posteriorly, the meshes were washed with three drops of distilled water and stained for 1 min with uranyl acetate (Sigma, St Louis, MO) at a concentration of 1%. The excess of uranyl acetate was drained on blotting paper, and the grid was dried during 10 min before image acquisition. Micrographs were taken with a transmission electron microscope Philips CM100 (Thermo Fisher Scientific, Hillsboro, USA) at an acceleration voltage of 80kV with a TVIPS TemCam-F416 digital camera (TVIPS GmbH, Gauting, Germany).

## Generation of the BeeComm_002 bacterial community

Thirteen bacterial isolates that represent the most abundant species found in the bee gut were grown individually and assembled together to generate a bacterial community named BeeComm_002 (see Key Resources Table). Each individual member was grown in agar plates of its respective preferential medium and culture conditions for 3 d, bacteria were harvested from the plates, diluted to $OD_{600}$ = 1 in PBS, mixed together in the same proportions, and placed in a glycerol stock (20% final concentration) at –80°C. *Gilliamella* strains were grown in BHIA plates and incubated in anaerobic conditions at 35°C. *Bifidobacterium*, *Lactobacillus,* and *Commensalibacter* strains grew anaerobically and at 35°C in Man, Rogosa, and Sharpe agar plates supplemented with fructose and L-cysteine. *S. alvi* was grown in Trypticase Soy agar plates, *B. apis* grown in Columbia Broth agar plates supplemented with 5% sheep blood, and both isolates were placed in micro-aerophilic conditions with 5% $CO_2$ and at 34°C.

## Co-colonization of bees with *F. perrara* and BeeComm_002

Each gene-deletion mutant, the wt and the *ihfA** mutant of *F. perrara* were grown as previously mentioned and diluted to $OD_{600} = 1$ on the day the colonization experiment was performed. Then, 100 µl of a given *F. perrara* genotype at $OD_{600} = 1$, 100 µl of BeeComm_002 at $OD_{600} = 1$ and 800 µl of a PBS and sugar water solution (1:1) were mixed together. As BeeComm_002 is composed of 13 isolates and was mixed in a 1–1 ratio with *F. perrara*, *F. perrara* was 13 times more abundant than any individual member of this synthetic community. For the control treatments where BeeComm_002 was not added (mono-associations), 100 µl of a given *F. perrara* genotype at $OD_{600} = 1$ were diluted in 900 µl of a PBS and sugar water solution (1:1). Gnotobiotic bees were generated, manipulated, and inoculated with bacteria as previously mentioned. Ileum/pylorus regions were dissected at day 10 after inoculation and scab formation was assessed before snap-freezing the samples in liquid nitrogen. Samples were placed at –80°C until DNA was extracted using previously mentioned methods. Bacterial abundance was quantified using the approach described in the section 'Bacterial quantification by qPCR.' Primers used in this experiment are found in Key Resources Table.

## Genome resequencing

All gene deletion strains and the three white variants of *F. perrara* PEB091 were subjected to Illumina genome resequencing at Novogene (2 × 150 bp). Mutations in the resequenced genomes relative to the genome of the wt strain of *F. perrara* were analyzed with *breseq* (**Deatherage and Barrick, 2014**). The detailed results of this analysis can be found at https://drive.switch.ch/index.php/s/kCNTp4g7n60ffMi.

## Statistical analysis

Statistical analysis was performed using GraphPad Prism v6.01 or R. Normality was analyzed using D'Agostino–Pearson and/or Shapiro–Wilk test. For comparison with two conditions, *t*-test was used for parametric data and Wilcoxon sum-rank test or Kolgomorov–Smirnov test for non-parametric data. For comparison of the growth curves of *F. perrara* wt and *ihfA**, we used a permutation test 'compareGrowthCruves' included in the statmod package of R (**Elso et al., 2004**). For the analysis of the CFU counts, a linear model with binomial distribution was calculated with CFUs as the dependent variable and Genotype and Experimental Replicate as independent variables. Post hoc comparisons between genotypes were performed using the emmeans R package (https://cran.r-project.org/web/packages/emmeans/index.html). For the comparison of the cell lengths, Wilcoxon sum-rank test was performed.

# Acknowledgements

We thank Stephan Gruber for input regarding the characterization of the ihfA* mutant strain and general feedback for the manuscript. We are grateful to Nicolas Neuschwander and Lucie Kesner for helping with some of the bee experiments. We thank Tania Trabajo for her help with the single-cell imaging. This project was funded by the ERC-StG 'MicroBeeOme' (grant agreement no. 714804), the Swiss National Science Foundation (grant agreement no. 31003A_160345 and 31003A_179487), the NCCR Microbiomes (all awarded to PE). JP acknowledges funding by the European Research Council (ERC) under the European Union's Horizon 2020 Research and Innovation Program (grant agreement no. 742739).

# Additional information

### Competing interests

Konstantin Schmidt: K. Schmidt is affiliated with Roche Diagnostics. The author has no financial interests to declare. The other authors declare that no competing interests exist.

## Funding

| Funder | Grant reference number | Author |
|---|---|---|
| European Research Council | 714804 | Philipp Engel<br>Théodora Steiner |
| Schweizerischer Nationalfonds zur Förderung der Wissenschaftlichen Forschung | 31003A_179487 | Philipp Engel<br>Yassine El Chazli<br>Konstantin Schmidt |
| Schweizerischer Nationalfonds zur Förderung der Wissenschaftlichen Forschung | 31003A_160345 | Philipp Engel<br>Olivier Emery |
| European Research Council | 742739 | Stefan Leopold-Messer<br>Joern Piel |
| Schweizerischer Nationalfonds zur Förderung der Wissenschaftlichen Forschung | 51NF40_180575 | Philipp Engel<br>Gonçalo Santos-Matos |

The funders had no role in study design, data collection and interpretation, or the decision to submit the work for publication.

### Author contributions

Konstantin Schmidt, Conceptualization, Data curation, Formal analysis, Validation, Investigation, Visualization, Methodology, Writing – original draft, Writing – review and editing; Gonçalo Santos-Matos, Conceptualization, Data curation, Formal analysis, Validation, Investigation, Visualization, Methodology, Writing – review and editing; Stefan Leopold-Messer, Conceptualization, Formal analysis, Investigation, Methodology, Writing – original draft, Writing – review and editing; Yassine El Chazli, Formal analysis, Investigation, Methodology, Writing – review and editing; Olivier Emery, Conceptualization, Data curation, Formal analysis, Investigation, Methodology, Writing – review and editing; Théodora Steiner, Investigation, Methodology; Joern Piel, Conceptualization, Supervision, Funding acquisition, Project administration, Writing – review and editing; Philipp Engel, Conceptualization, Supervision, Funding acquisition, Investigation, Visualization, Writing – original draft, Project administration, Writing – review and editing

### Author ORCIDs

Gonçalo Santos-Matos http://orcid.org/0000-0001-8303-6735
Stefan Leopold-Messer http://orcid.org/0000-0003-3490-3550
Philipp Engel http://orcid.org/0000-0002-4678-6200

### Decision letter and Author response

Decision letter https://doi.org/10.7554/eLife.76182.sa1
Author response https://doi.org/10.7554/eLife.76182.sa2

# Additional files

### Supplementary files
• Transparent reporting form

### Data availability

RNA sequence datasets are available under NCBI Gene Expression Omnibus ID GSE189728. Code, scripts and numeric data files of experimental data have been deposited on Zenodo: https://zenodo.org/record/7778751#.ZCMOguxBw0p.

The following datasets were generated:

| Author(s) | Year | Dataset title | Dataset URL | Database and Identifier |
|---|---|---|---|---|
| Schmid K, Santos-Matos G, Leopold-Messer S, El-Chazli Y, Emery O, Steiner T, Piel J, Engel P | 2023 | Integration host factor regulates colonization factors in the bee gut symbiont Frischella perrara | https://doi.org/10.5281/zenodo.7778751 | Zenodo, 10.5281/zenodo.7778751 |
| Engel P, Schmidt K | 2021 | RNA sequencing of Frischella perrara | https://www.ncbi.nlm.nih.gov/geo/query/acc.cgi?acc=GSE189728 | NCBI Gene Expression Omnibus, GSE189728 |

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
