## [Editor Report]

This fundamental work substantially advances our understanding of the genetic basis of how a very prevalent bee symbiont, Frischella perrara, colonizes the gut of these insects, by identifying novel players in this process and raising new questions related to their mode of action. The authors characterized spontaneous mutants in an important regulator, and showed that this regulator controls the expression of several genes required for gut colonization, they constructed deletion mutants on these genes and characterized these mutants both in vitro and in colonization assays in the presence and absence of other gut symbionts, and provide insights on the mode of action of the novel players identified during host colonization. The combination of approaches used is exceptional and established new standards in the field of host-microbe interactions aiming to understand the molecular players involved in the colonization of gut symbionts.

---

## [Decision Letter]

**Decision letter after peer review:**

Thank you for submitting your article "IHF regulates host colonization factors in the bee gut symbiont *Frischella perrara*" for consideration by *eLife*. Your article has been reviewed by 3 peer reviewers, including Karina B Xavier as Reviewing Editor and Reviewer #1, and the evaluation has been overseen by Gisela Storz as the Senior Editor. The following individuals involved in review of your submission have agreed to reveal their identity: François Leulier (Reviewer #3); Laura Florez Platino (Reviewer #4).

The reviewers have discussed their reviews with one another, and the Reviewing Editor has drafted this letter to help you prepare a revised submission.

In general, all three reviewers thought the work is solid and that the conclusions are well supported by the data. However, there was also a general consensus that the manuscript would benefit from additional experiments and context to clarify and increase the biological relevance of the mechanisms identified in this study towards a better understanding of the importance of the genetic factors identified here in symbiont gut colonization.

Below is a list of suggestions for essential revisions proposed by the reviewers after the consultation session, that we hope you can use to improve the manuscript. Additionally, you can also find in the end of this letter the sperate "recommendations for the authors" written by each of the reviewers which should help to improve the manuscript and to understand the rationale of each of the reviewers. You do not need to reply to each of the individual comments from "recommendations for the authors", for submission of a revised version you only need to reply to the essential revision’s points.

Essential revisions:

We think that the manuscript requires additional experiments aiming to address the biological relevance of the genetic factors identified here. Namely, the paper requires a more detailed characterization of the mechanisms underlying the colonization phenotypes of the mutants identified here on both the host and bacterial sides to better understand the relevance of these processes on host biology and/or bacterial physiology. The points below are suggestions for experiments to achieve this aim, it is possible that there are other phenotypes/experiments beyond the ones suggested here that can also be used, or that could be even better. to achieve this goal.

1) The mutants were studied in mono-colonized bees, we suggest that colonization phenotypes of some the mutants (from Figure 6) are also tested in bees colonized with a complex community, as the results might be different and should help to further understand the role of the mechanisms affected in these mutants in the bee gut. For example, we think it would be interesting to test the mutants related to colibactin and T6SS.

2) The manuscript would also improve with an additional characterization of the mutants in terms of growth rate and cell shape, or other characterizations of the physiology of these bacterial mutants, to help understand why some of the mutants are compromised in their ability to colonize the bee gut.

3) Given the importance of understanding the mechanisms that are required for the occupancy of distinct physical niches. We suggest that the authors test if there is any difference in the spatial gut colonization of the mutates studied here. The data in Figure 6 refers to colonization in the pylorus and ileum. We suggest that these results are compared with results with the CFUs recovered from the whole gut to see if any of these mutants has a different spatial colonization niche.

4) The importance of the scab phenotype is highlighted in the manuscript (although additional context is need (see point 6)), one of the mutants studied colonizes at high loads but is impaired in inducing the scab phenotype. Additional characterization of the impact of this mutant for the host could reveal additional insight related to the impact of the scab phenotype on host performance or health.

5) Technical issues: i) The number of bees tested in Figure 3 should be the same in the two time points. ii) clarify if some of the results might have been affected by differences in OD and cell numbers when comparing the WT and the different mutants given that some mutants are affected in cell size/shape. Calibrations between might OD and cell concentrations might be required to ensure that there was no bias in any of the experiments in which same OD was assumed as equivalent titers of the mutant and the wt.

6) There are some points that can be improved with additional clarification/re-writing: i) The authors should include in the manuscript the motivation for the choice of F. perrara. What is known about the relevance of this bacterium for the host, how common/prevalent is this symbiont in natural conditions? ii) the manuscript requires additional information about the relevance of the scab phenotype in symbiosis and/or pathogenesis, what is known about the relevance of these phenotype for the host. The importance of the scab phenotype should be clear already in the abstract (iii) the abstract ends with the conclusion that the different mutants identified had different colonization defects, this is a very vague conclusion, it would be better to try to be more specific and explained for example what was learnt with the Type VI SS mutants, or include additional conclusion from the experiments suggested above.

*Reviewer #1 (Recommendations for the authors):*

1. As the authors emphasize, one important question in this field is the understanding of the mechanisms that are required for the occupancy of distinct physical niches. Therefore, I think it would be interesting to determine the spatial gut colonization of the mutates studied here. The data in Figure 6 refers to colonization in the pylorus and ileum. I think it would be interesting to compare these results with the CFUs recovered from the whole gut to see if there is any of these mutants that has a different spatial colonization niche.

2. Mutants involved in the colibactin production were only mildly affected in gut colonization, can it be that colibactin is not important in mono-colonization conditions, but it is required for colonization in the presence of the other gut symbionts? This hypothesis should be considered/discussed.

3. Throughout the paper it is often written that the results show that IHF plays an important role in gut colonization. I think it is clear that the effect of IHF in gut colonization is related to its effect of the expression of several colonization such as the pili genes. Therefore, I think it would be better to emphasize that the results presented here show that IHF acts as major regulator of the expression of factors involved in gut colonization. It is indeed still possible that IHF also has a direct role in colonization, so I think it still makes sense to raise that possibility in the discussion (as it is written in lines 328), but the major conclusion should be that the results indicate that IHF is acting as a regulator of the expression of factors which are important in colonization.

*Reviewer #2 (Recommendations for the authors):*

Overall, the study is thorough and the main conclusions supported by the results however this reviewer feels that the study deserve further insights to reach the level of advances expected by this referee for an *eLife* manuscript, however I let the editor decide on this parameter. The study identifies new genetic factors that support symbiont colonization in the bee gut but so far falls short at dissecting the mechanisms underlying the phenotypes on both the host and bacterial sides, which I believe would be necessary for publication in *eLife*. As it is, my feeling is that the manuscript in its current form would be a better fit for generalistic microbiology journal such as mBio.

*Reviewer #3 (Recommendations for the authors):*

– Title: the abbreviation IHF is unlikely to relate to a broad audience. I suggest modifying.

– Lines 49-50: this can be interpreted as if only bacterial factors are involved in determining spatial distribution in the honeybee gut. I suggest mentioning that host factors are potentially involved as well, unless there is opposing evidence.

– Lines 68-72: I suggest to also include some words on the occurrence of the scab phenotype as has been described previously (Engel et al. 2015 mBio), to give the reader a better sense of the potential relevance of this phenotype. Also, here or elsewhere it is worth clarifying that this is not associated to disease or other negative impact on the host, if that is the case.

– Lines 101-105: please revise the text or figure, as the amino acids or positions referred to do not match (See specific comment on Figure 1).

– Lines 148-150: How did the authors make sure that the colonies that were plated are indeed F. perrara? Is having contamination in the feed for the bees or while plating the guts unlikely? Please clarify here or in the corresponding methods section.

– Lines 168-170: given that a difference in cell shape and size is reported for the wt and the mutant, it would have been relevant to make a calibration of OD vs cell concentration. It is likely that this underlies the difference in CFUs noted here.

Therefore, I find it relevant to expand on this point, providing a quantitative estimate of this difference and making sure that this does not generate a bias in any of the experiments in which same OD was assumed as equivalent titers of the mutant and the wt.

– Line 248-250: please revise the grammar of this sentence. Also, it is not fully clear why this was not unexpected. I suggest rephrasing/clarifying.

– Line 368: please provide a reference for the claim that no other honeybee symbiont causes the scab phenotype.

– Lines 100-101: as far as I am aware, it is only present in Gram-negative bacteria. I suggest to state more precisely.

– Line 434: I suggest indicating directly here at least if this was via antibiotic treatment (and which).

– Lines 439-440: See specific comment for lines 168-169.

– Line 657: please revise sentence.

– Figure 1: Note that in C and in the legend there seems to be a mislabeling or misplacement of the mutations in the sequence. Lysine > Serine is written but on the sequence the green rectangle indicates Leucine (L) > Methionine (M). Also for the Proline mutations, the highlighted region of the sequence alignment does not match what is indicated in the text, both in terms of amino acids and location.

– I believe that the manuscript would benefit from including the following points in the discussion (or elsewhere), even if there are no conclusive answers to some of the points:

– How might the presence of other bacteria affect F. perrara colonization (in relation to the genes in focus) and the scab phenotype? Are the titers of F. perrara in the experiments with gnotobiotic bees comparable to titers of F. perrara under natural conditions?

– Are different strains of F. perrara known in honey bees, how consistently do specific strains colonize within or across different colonies? How representative is the strain used in this study?

---

## [Author Response]

Essential revisions:We think that the manuscript requires additional experiments aiming to address the biological relevance of the genetic factors identified here. Namely, the paper requires a more detailed characterization of the mechanisms underlying the colonization phenotypes of the mutants identified here on both the host and bacterial sides to better understand the relevance of these processes on host biology and/or bacterial physiology. The points below are suggestions for experiments to achieve this aim, it is possible that there are other phenotypes/experiments beyond the ones suggested here that can also be used, or that could be even better. to achieve this goal.

Thank you for this clear summary and the suggestions. We have conducted a more detailed characterization of the mutants in the revised manuscript. In some cases, we have carried out the experiments as suggested by the reviewers. In other cases, we have come up with a different approach that we believe addresses the main point of the critique, but from a more interesting perspective.

1) The mutants were studied in mono-colonized bees, we suggest that colonization phenotypes of some the mutants (from Figure 6) are also tested in bees colonized with a complex community, as the results might be different and should help to further understand the role of the mechanisms affected in these mutants in the bee gut. For example, we think it would be interesting to test the mutants related to colibactin and T6SS.

During the revision, we tested the colonization of the *Frischella perrara* mutants in the presence and absence of a complex community containing 13 strains representing the core members of the bee gut microbiota (BeeComm_002). In short, we colonized bees with a 1:1 mix of *Frischella perrara* and the BeeComm_002 (i.e., *F. perrara* was present 13x more concentrated than any of the individual strains) and analysed the colonization levels of *Frischella perrara* using qPCR at 10 days post-colonization (Figure 7 and Figure 7 —figure supplement 1). We tested all eight *Frischella perrara* genotypes (i.e. wt, *ihfA**, and the six inframe deletion mutants: three T6SS mutants, colibactin mutant, aryl polyene mutant, and T4 pili mutant) in the presence and in the absence of the BeeComm_002. We carried out two independent replicates of this experiment. A total of 20 bees were analysed for each treatment. Interestingly, while the colonization of the wt strain of *F. perrara* was not affected, three of the six tested in-frame deletion mutants showed an increased colonization defect in the presence the BeeComm_002 (Figure 7). This included two of the T6SS mutants and the mutant of the aryl polyene. Moreover, the presence of the BeeComm_002 led to a reduction of the number of bees colonized and number of scabs formed (Figure 7 —figure supplement 1). This trend was seen for most strains tested but not for the wt.

We have included these results into the revised manuscript, by expanding the Results section with a new chapter on the role of the colonization factors in the presence of a community reflecting the conventional microbiota of bees.

2) The manuscript would also improve with an additional characterization of the mutants in terms of growth rate and cell shape, or other characterizations of the physiology of these bacterial mutants, to help understand why some of the mutants are compromised in their ability to colonize the bee gut.

We have carried out the following additional analyses to better understand why some mutants do not colonize the gut anymore or do not induce the scab phenotype:

i) We carried out electron microscopy on some of the genotypes which showed that cells of the wt but not the *ΔpilE* mutant have pili-like cell protrusions. This confirms that the *ΔpilE* mutant has an effect of pili formation which may be key for host attachment in the pylorus. Corresponding images have been added to the Supplementary material as Figure 6 —figure supplement 3.

ii) We carried out FISH microscopy on guts colonized with all eight strains to assess if those mutants which still colonize the host are also able to localize to the epithelial surface of the pylorus where the wt induces the scab phenotype. As expected from the CFU counts and the absence of pili structures, no bacteria were found to be attached to the host epithelium in the cross-sections of bees colonized with the *ihfA** and *ΔpilE* mutant. In contrast, the two T6SS mutants which still colonized at high levels (*Δhcp2* and *Δhcp1/hcp2*), but which did not cause the scab phenotype anymore, both still colonized the epithelial surface of the pylorus in a very similar way as the wt. This is intriguing as it suggests that the absence of the scab phenotype in these two mutants is not due to the inability of the bacteria to attach to the epithelium (where the scab forms). It confirms our previous speculation that one of the T6SS may trigger scab formation by e.g. delivering a host-targeting effector. These results have been added to the manuscript in the second to last results chapter and the FISH images added to Figure 6B (also, Figure 6 —figure supplements 6 and 7).

iii) We also characterized the in vitro growth and viability of the mutant strains in more detail. We prepared bacterial solutions at the same optical density – OD 0.1 – and plated serial dilutions on semi-solid agar to calculate the number of colony forming units (CFUs) (Figure 6 —figure supplement 4). We observed that two mutants – *Δhcp1/hcp2* and *ΔpilE* – had less CFU counts than the wt. However, we do not think these differences affected the capacity to colonize. In the mono-association experiment, the *Δhcp1/hcp2* mutant colonized to the same loads as *Δhcp2* and *ΔclbB*, whose loads did not differ from the wt in the OD to CFU experiment. Moreover, while *ΔpilE* had a severe colonization defect in mono-association, it had higher loads than the *Δhcp1/hcp2* in the OD to CFU experiment. Therefore, having reduced CFU counts in the OD to CFU experiment does not correlate with the capacity to colonize. Moreover, we characterized the growth of the six gene-frame deletion mutants, *ihfA** and wt strains in liquid BHI medium, under anaerobic conditions, and saw that all strains grow similarly to the wt (Figure 6 —figure supplement 2). This shows that these phenotypical characteristics observed in vitro may tell us little about the ability of strains to colonize the gut. Nevertheless, we have added the growth curves into the Supplementary material.

iv) We obtained light microscopy images for the six gene-frame deletion mutants, *ihfA** and wt strains and measured cell length. No differences were found in this experiment (Figure 6 —figure supplement 3), in opposition to the experiment shown in Figure 1 —figure supplement 2. This new experiment was done in BHI, which is the liquid medium we generally use to grow Frischella. In contrast, the initial experiment was done in TYG medium, because this was the condition in which the spontaneous ihfA/B mutants were isolated.

3) Given the importance of understanding the mechanisms that are required for the occupancy of distinct physical niches. We suggest that the authors test if there is any difference in the spatial gut colonization of the mutates studied here. The data in Figure 6 refers to colonization in the pylorus and ileum. We suggest that these results are compared with results with the CFUs recovered from the whole gut to see if any of these mutants has a different spatial colonization niche.

We carefully considered this suggestion, and decided to focus on comparing the spatial distribution of the different gene-deletion mutants within the pylorus instead of across gut compartments. We did so for two reasons: (i) The pylorus is the preferential niche of this bacterium in conventional bees, and only there it forms a biofilm attached to the host. Studying the spatial distributions of the different mutants within the pylorus could help to identify mechanisms that regulate biofilm formation in this gut compartment. Moreover, comparing the spatial distribution of these genotypes may lead to a better understanding of the scab formation, namely by studying those mutants that colonize but do not induce the scab. (ii) The disadvantage of mono-association experiments in the laboratory is that there is no competition with other bacteria, and hence niches in other gut compartments are available for invasion even though they would never be colonized by the bacterium of interest in conventional bees. So, instead of looking at the CFU levels in other parts of the gut, we decided to specifically look at the characteristic colonization pattern of the different genedeletion mutants in the pylorus (Figure 6B and Figure 6 —figure supplement 6 and 7). See also reply to point 1. This allowed us to assess which mutants are still able to adhere to the epithelial surface and whether mutants that show a defect in scab formation, but not colonization, may have a defect in adhesion and biofilm formation in the regions where scab material usually accumulates. Strikingly, this was not the case. All mutants for which we observed high levels of colonization still colonized the crypts of the epithelial surface of the pylorus region, even the T6SSs mutants (*Δhcp2* and *Δhcp1/hcp2*) which did not induce the scab phenotype anymore. This is interesting, as it suggests that these mutants are still able to interact with the host, yet are not capable of triggering the scab anymore. It is possible that the mutated T6SS targets the host or delivers effector proteins that lead to the built up of the scab material. This adds exciting new data to the current manuscript and opens the door to further experimental approaches to describe the function of these T6SSs in more detail in the future. The FISH images have been added to Figure 6B (but also Figure 6 —figure supplement 6 and 7).

4) The importance of the scab phenotype is highlighted in the manuscript (although additional context is need (see point 6)), one of the mutants studied colonizes at high loads but is impaired in inducing the scab phenotype. Additional characterization of the impact of this mutant for the host could reveal additional insight related to the impact of the scab phenotype on host performance or health.

Please see our reply to point 3. The FISH microscopy shows that the spatial colonization of this mutant, *Δhcp2*, is similar to the *ΔclbB* mutant that has similar loads and induces scab formation. We now believe there are specific effector proteins that are necessary to trigger the scab and started a project in the lab to identify them. This said, we don’t know yet if the scab phenotype has an impact on the host. However, we know that it is widely distributed in bees of the species *Apis mellifera* (you can find it in 25-80% of all adult bees in every colony) and that it is specifically associated with the presence of *F. perrara.* Other bee species, although they harbour similar bacteria in the gut have not been reported to exhibit this phenotype. This makes this phenotype really interesting to study to understand fundamental aspects of the specificity of microbiota-host interactions, even though we do not know yet if this phenotype has any relevance for the fitness of the host.

Nevertheless, we know that *F. perrara* triggers the host immune response and that this very likely leads to scab formation. This activation may protect the honey bee from opportunistic pathogens, creating a filtering mechanism in the gut. We will explore this hypothesis in the future.

5) Technical issues: i) The number of bees tested in Figure 3 should be the same in the two time points.

Bees were sampled from the same cages at the two different timepoints. As sampling at Day 5 was an intermediate timepoint we removed a smaller number of bees to ensure to have enough bees left for the later timepoint. As day 10 was the final timepoint, we analysed all bees that were left. We believe it would be a pity to discard data to make the two groups equal. Also, we carried out a proper time course in a follow-up experiment (see Figure 3) to compare across timepoints. Therefore, we believe that these differences have no influence on the overall point we want to make with these results.

ii) clarify if some of the results might have been affected by differences in OD and cell numbers when comparing the WT and the different mutants given that some mutants are affected in cell size/shape. Calibrations between might OD and cell concentrations might be required to ensure that there was no bias in any of the experiments in which same OD was assumed as equivalent titers of the mutant and the wt.

Both OD and CFUs have caveats when measuring bacterial cell number in the inocula. As pointed out by the reviewers, OD is affected by cell size and shape. Less often considered, but as important is the fact that bacterial cells can clump together and form a single colony instead of several separate ones. This will influence CFU enumerations. Therefore, calibrating one with the other may not be ideal. However, we can exclude that small differences in the actual number of cells in the inocula influenced our colonization results. (i) In the time course experiment, the total number of viable cells (i.e. forming CFUs) in the inocula of the *ihfA** mutant was much higher than in the wt (Figure 3D). Yet this strain poorly colonized the bees. (ii) To correspond OD to CFU, we diluted all genotypes to the same optical density (OD 0.1) and quantified the number of colony forming units in these bacterial solutions (Figure 6 —figure supplement 4). The *ΔpilE* and *Δhcp1/hcp2* had less CFU counts than the wt. However, we do not think these affected the capacity to colonize. In the mono association experiment, the *Δhcp1/hcp2* mutant colonized to the same loads as *Δhcp2* and *ΔclbB*, whose loads did not differ from the wt in the OD to CFU experiment. Moreover, *ΔpilE* had a severe colonization defect in mono-association, while it had higher loads than the *Δhcp1/hcp2* in the OD to CFU experiment. Finally, we have done an experiment in which we measured cell length (see also reply IV to point 2) and did not observe significant differences between strains. This was done in BHI, whereas the initial experiment was done in TYG medium. We did not see any association between cell length and colonization capacity in this new experiment.

6) There are some points that can be improved with additional clarification/re-writing: i) The authors should include in the manuscript the motivation for the choice of F. perrara. What is known about the relevance of this bacterium for the host, how common/prevalent is this symbiont in natural conditions? ii) the manuscript requires additional information about the relevance of the scab phenotype in symbiosis and/or pathogenesis, what is known about the relevance of these phenotype for the host.

We have included a corresponding section in the introduction, which reads as follows:

“F. perrara is highly prevalent across worker bees and colonies of *A. mellifera*, and related bacteria have also been found in Apis cerana. Moreover, between 25%-80% of all worker bees of a colony harbor a visible scab phenotype in the pylorus region of the gut, which has been shown to strongly correlate with a high abundance of F. perrara (34). However, the impact of these phenotype on the host has remained elusive.”

We have also changed the abstract. See Reply to (iii).

The importance of the scab phenotype should be clear already in the abstract (iii) the abstract ends with the conclusion that the different mutants identified had different colonization defects, this is a very vague conclusion, it would be better to try to be more specific and explained for example what was learnt with the Type VI SS mutants, or include additional conclusion from the experiments suggested above.

We changed the abstract so to be more specific about our findings. We now highlight the potential role of the T6SS in causing the scab phenotype. The abstract reads as follows:

“Bacteria colonize specific niches in the animal gut. However, the genetic basis of these associations is often unclear. The proteobacterium Frischella perrara is a widely distributed gut symbiont of honey bees. It colonizes a specific niche in the hindgut and causes a characteristic melanization response. Genetic determinants required for the establishment of this association, or its relevance for the host, are unknown. Here, we independently isolated three point mutations in genes encoding the DNA-binding protein integration host factor (IHF) in F. perrara. These mutants abolished the production of an aryl polyene metabolite causing the yellow colony morphotype of F. perrara. Inoculation of microbiota-free bees with one of the mutants drastically decreased gut colonization of F. perrara. Using RNAseq we found that IHF affects the expression of potential colonization factors, including genes for adhesion (Type 4 pili), interbacterial competition (Type 6 secretion systems), and secondary metabolite production (colibactin and aryl polyene biosynthesis). Gene deletions of these components revealed different colonization defects depending on the presence of other bee gut bacteria. Interestingly, one of the T6SS mutants did not induce the scab phenotype anymore, despite colonizing at high levels, suggesting an unexpected role in bacteria-host interaction. IHF is conserved across many bacteria and may also regulate host colonization in other animal symbionts.”

Reviewer #1 (Recommendations for the authors):1. As the authors emphasize, one important question in this field is the understanding of the mechanisms that are required for the occupancy of distinct physical niches. Therefore, I think it would be interesting to determine the spatial gut colonization of the mutates studied here. The data in Figure 6 refers to colonization in the pylorus and ileum. I think it would be interesting to compare these results with the CFUs recovered from the whole gut to see if there is any of these mutants that has a different spatial colonization niche.

See our reply to point 3 of Essential Revisions. But briefly, we opted to look at the spatial occupation of the gene-deletion mutants within the pylorus, the natural niche of this bacterium, using FISH microscopy (Figure 6B and Figure 6 —figure supplement 6 and 7). The mutants that colonize but do not induce the scab occupy the same locations as those mutants that lead to scab formation.

2. Mutants involved in the colibactin production were only mildly affected in gut colonization, can it be that colibactin is not important in mono-colonization conditions, but it is required for colonization in the presence of the other gut symbionts? This hypothesis should be considered/discussed.

As outlined in point 1 of Essential Revisions, we have tested the ability of all mutants to colonize the bee gut in the presence of a complex bacterial community (Figure 7 and Figure 7 —figure supplement 1). There was no statistically significant difference in colonization levels of the colibactin mutant between mono-association and the presence of the community. However, less bees were colonized by the colibactin mutant when the complex bacterial community was present (Figure 7 —figure supplement 1B).

Therefore, it is possible that colibactin plays a role in the presence of competitors although to a lesser extent than other genes tested (e.g: *Δhcp2*). Future experiments with more replicates and fewer *F. perrara* cells as compared to the complex bacterial community (here we used a 1:1 ratio) may help to identify a clearer phenotype of the colibactin mutant.

3. Throughout the paper it is often written that the results show that IHF plays an important role in gut colonization. I think it is clear that the effect of IHF in gut colonization is related to its effect of the expression of several colonization such as the pili genes. Therefore, I think it would be better to emphasize that the results presented here show that IHF acts as major regulator of the expression of factors involved in gut colonization. It is indeed still possible that IHF also has a direct role in colonization, so I think it still makes sense to raise that possibility in the discussion (as it is written in lines 328), but the major conclusion should be that the results indicate that IHF is acting as a regulator of the expression of factors which are important in colonization.

We agree that IHF impacts colonization by regulating genes that help *F. perrara* to interact with its host environment. However, we argue that therefore IHF itself can be considered a colonization factor. We should keep in mind that we do not know yet how the genes regulated by IHF influence colonization. As in the case of IHF, the effect does not need to be direct, yet we consider them colonization factors, as we see a change in colonization when we delete them.

Reviewer #2 (Recommendations for the authors):Overall, the study is thorough and the main conclusions supported by the results however this reviewer feels that the study deserve further insights to reach the level of advances expected by this referee for an eLife manuscript, however I let the editor decide on this parameter. The study identifies new genetic factors that support symbiont colonization in the bee gut but so far falls short at dissecting the mechanisms underlying the phenotypes on both the host and bacterial sides, which I believe would be necessary for publication in eLife. As it is, my feeling is that the manuscript in its current form would be a better fit for generalistic microbiology journal such as mBio.

We believe that with the additional results added to this manuscript, the study becomes even more interesting for a broad audience and bolsters some of the previous findings.

Reviewer #3 (Recommendations for the authors):– Title: the abbreviation IHF is unlikely to relate to a broad audience. I suggest modifying.We suggest to change it to: “Integration host factor regulates colonization factors in the bee gut symbiont Frischella perrara”– Lines 49-50: this can be interpreted as if only bacterial factors are involved in determining spatial distribution in the honeybee gut. I suggest mentioning that host factors are potentially involved as well, unless there is opposing evidence.

This has been changed.

– Lines 68-72: I suggest to also include some words on the occurrence of the scab phenotype as has been described previously (Engel et al. 2015 mBio), to give the reader a better sense of the potential relevance of this phenotype. Also, here or elsewhere it is worth clarifying that this is not associated to disease or other negative impact on the host, if that is the case.

See our reply to the Essential Revisions.

– Lines 101-105: please revise the text or figure, as the amino acids or positions referred to do not match (See specific comment on Figure 1).

We have changed the legend of the figure so that it becomes clear what the sequence alignment shows. See our specific reply to comment about Figure 1.

– Lines 148-150: How did the authors make sure that the colonies that were plated are indeed F. perrara? Is having contamination in the feed for the bees or while plating the guts unlikely? Please clarify here or in the corresponding methods section.

Microbiota-free bees that are not colonized serve as control and in such cases, we rarely get bacteria grown on the growth medium used to cultivate Frischella. Frischella makes quite characteristic colonies with a yellow color, which take 3-4 days to develop. Colonies of a contamination would be quite obvious and have indeed been observed in the past, but not in any of the experiment presented here. We also checked for possible contaminations in the bees. We do so by sampling a subset of bees one day before *F. perrara* inoculation.

– Lines 168-170: given that a difference in cell shape and size is reported for the wt and the mutant, it would have been relevant to make a calibration of OD vs cell concentration. It is likely that this underlies the difference in CFUs noted here. Therefore, I find it relevant to expand on this point, providing a quantitative estimate of this difference and making sure that this does not generate a bias in any of the experiments in which same OD was assumed as equivalent titers of the mutant and the wt.

See our reply to the Essential Revisions. But briefly, no differences were seen between wt and *ihfA** when measuring CFU counts of bacterial solutions at OD_600_=0.1 (Figure 6 —figure supplement 4).

– Line 248-250: please revise the grammar of this sentence. Also, it is not fully clear why this was not unexpected. I suggest rephrasing/clarifying.

This has been corrected.

– Line 368: please provide a reference for the claim that no other honeybee symbiont causes the scab phenotype.

This has been corrected.

– Lines 100-101: as far as I am aware, it is only present in Gram-negative bacteria. I suggest to state more precisely.

We are not sure to which sentence this refers.

– Line 434: I suggest indicating directly here at least if this was via antibiotic treatment (and which).

No AB treatment was used to generate MF bees. Pupae were collected from brood frames taken from the hives, broods were surface sterilized, pupae were extracted and maintained in an isolated incubator for three days before each experiment. Additionally, controls assessing for possible contaminations were performed the day before the experiment by sampling different individuals.

– Lines 439-440: See specific comment for lines 168-169.

See our reply to the Essential Revisions.

– Line 657: please revise sentence.

This has been corrected.

– Figure 1: Note that in C and in the legend there seems to be a mislabeling or misplacement of the mutations in the sequence. Lysine > Serine is written but on the sequence the green rectangle indicates Leucine (L) > Methionine (M). Also for the Proline mutations, the highlighted region of the sequence alignment does not match what is indicated in the text, both in terms of amino acids and location.

The alignment does not show the wt and the mutated variants, but the homologs *ihfA* and *ihfB* in *F. perrara* and *E. coli*. The positions on top of the plot do not match with the position of each individual sequence because there are gaps in the alignment. We have changed the legend of Figure 1C so that it becomes clear what is shown in this Figure panel.

– I believe that the manuscript would benefit from including the following points in the discussion (or elsewhere), even if there are no conclusive answers to some of the points:– How might the presence of other bacteria affect F. perrara colonization (in relation to the genes in focus) and the scab phenotype? Are the titers of F. perrara in the experiments with gnotobiotic bees comparable to titers of F. perrara under natural conditions?– Are different strains of F. perrara known in honey bees, how consistently do specific strains colonize within or across different colonies? How representative is the strain used in this study?

We tried to keep the discussion focused on the results. Therefore, we included several discussion points about the presence of the BeeComm_002. We thought the other two points were less relevant for the discussion. However, to answer these questions: Yes, the loads are comparable to those present in conventional bees. See e.g., Engel et al. mBio, or Kesnerova et al. ISME J. We have isolated different strains of *F. perrara*. This bacterium is present in almost every bee, but only in bees that have high levels of the bacterium the scab is induced, usually in about 25-80% of all bees of a colony. We have added this info in the introduction. Moreover, we have done experiments with another *F. perrara* strain, isolated from a different continent than that here studied, and it induced the scab. This result suggest that scab formation is conserved and therefore the strain here studied is a good representative of *F. perrara.*